# Integrated assessment of future potential global change scenarios and their hydrological impacts in coastal aquifers. A new tool to analyse management alternatives in the Plana Oropesa-Torreblanca aquifer.

David Pulido-Velazquez[1, 2], Arianna Renau-Pruñonosa[3], Carlos Llopis-Albert[4], Ignacio Morell[3], Antonio-Juan Collados-Lara[1], Javier Senent-Aparicio[2], Leticia Baena-Ruiz[1]

[1]Instituto Geológico y Minero de España, Granada, Spain

[2]Universidad Católica de Murcia, Murcia, Spain

[3]Jaume I University, Castellón, Spain

[4]Universitat Politècnica de València, Valencia, Spain

*Correspondence to:* Antonio-Juan Collados-Lara (ajcollados@gmail.com)

**Abstract.** Any change in the components of the water balance in a coastal aquifer, whether natural or anthropogenic, can alter the fresh water-salt water equilibrium. In this sense Climate change (CC) and Land Use and Land Cover (LULC) change might significantly influence the availability of groundwater resources in the future. These coastal systems demand an integrated analysis of quantity and quality issues to obtain an appropriate assessment of hydrological impacts using density-dependent flow solutions. The aim of this work is to perform an integrated analysis of future potential global change (GC) scenarios and their hydrological impacts in a coastal aquifer, the Plana Oropesa-Torreblanca aquifer. It is a Mediterranean aquifer that extends over a 75 km$^2$ in which important historical LULC changes have been produced and are planned for the future. Future CC scenarios will be defined by using equi-feasible and non- feasible ensemble of projections based on the results of a multi-criteria analysis of the series generated from several Regional Climatic Models with different downscaling approaches. The hydrological impacts of these CC scenarios combined with future LULC scenarios will be assessed with a chain of models defined by a sequential coupling of rainfall-recharge models, crop irrigations requirements and irrigation returns models (for the aquifer and its neighbours that feeds it), and a density dependent aquifer approach. This chain of models, calibrated using the available historical data, allow testing the conceptual approximation of the aquifer behavior. They are also fed with series representatives of potential global change scenarios in order to perform a sensitivity analysis regarding future scenarios of rainfall recharge, lateral flows coming from the hydraulically-connected neighbouring aquifer, agricultural recharge (taking into account expected future LULC changes) and Sea Level Rise (SLR). The proposed analysis is valuable to improve our knowledge about the aquifer and so comprise a tool to design sustainable adaptation management strategies taking into account the uncertainty in future GC conditions and their impacts. The results show that GC scenarios produce significant increase in the variability of flow budget components and in the salinity.

## 1. Introduction

Certain coastal regions simultaneously suffer scarce surface water resources and significant water demand. As a result, the reliability of supplying the demand depends on groundwater resources, which therefore play an important role in the management of these systems (Sola et al., 2013; Renau et al., 2016). The analysis of coastal aquifer management problems is an important and complex issue in which water quantity and quality have to be considered together to predict the salinization

process, which depend on aquifer stratigraphy and other hydrodynamic factors (precipitation regime, tides, wave setup and storm surges, etc.) (Vallejos et al., 2015). Due to the interaction between freshwater and seawater, coastal aquifers have important hydrodynamic and hydrogeochemical peculiarities (Custodio, 2010). Any change in the components of the water balance can modify the fresh water-salt water equilibrium that defines the seawater intrusion processes (Yechieli and Sivan, 2011; Arslan and Demir, 2013). In the future, the difficulty of meeting demand in coastal systems will increase due to the impact of GC, which will reduce freshwater recharge, raise sea level and increase irrigation demand (Fujinawa, 2011; Unsal et al., 2014). Therefore, GC impacts will challenge the current water supply management of coastal aquifer (Rasmussen et al., 2013).

In recent years the number of studies of CC impacts focus on aquifers has grown fast (Green et al., 2011; Molina et al., 2013). A few of these studies considers quality impacts, though groundwater quality can be affected by GC in many different ways (Pulido-Velazquez et al., 2014; Dragoni and Sukhija, 2008). Only some of them are focused on the impacts of GC on coastal aquifers (Yechieli et al., 2010). Though coastal aquifers warrant greater attention since they are more vulnerable to GC (due to their connection with the sea and the interaction between fresh water and seawater) they have not been extensively studied (Rasmussen et al., 2013). To assess the salinization process and possible adaptation strategies properly, coastal aquifers require that water quantity and quality issues are analysed in an integrated way.

In order to analyse the potential impacts of future scenarios of CC on any hydrological system, we need to generate time series of climate variables and to use them as inputs of previously calibrated hydrological models with them.

In coastal aquifers there is another important issue that needs to be considered when analysing CC impacts, namely any change in sea level (Ketabchi et al., 2016;). A number of authors have studied the impacts of SLR in various coastal aquifers. Chan et al. (2011) showed that in a synthetic confined coastal aquifer in which recharge is steady there is no long-term impact on seawater intrusion. Werner and Simmons (2009) showed that in unconfined aquifers the influence of the inland boundary condition can be significant to its sensitivity to SLR. Rasmussen et al. (2013) analysed an inland coastal aquifer and found that minor SLRs did not seem to affect seawater intrusion as much. There are also reports by the IPCC (Church et al., 2013) and the European Environment Agency (EAA 2014), which focus on the analysis of historical and potential future SLR scenarios.

In addition, urban and agricultural development forces appropriate management rules to be applied if groundwater resources under different LULC scenarios are to be exploited sustainably (Robins et al., 1999; Grundmann et al., 2012;). A certain degree of overpumping usually occurs in Mediterranean coastal aquifers, particularly in summer, which encourages salinization processes (Rosenthal et al., 1992). This could be exacerbated by the future LULC scenarios.

Few studies have been published that analyse GC in an integrated way –along with its impacts on future LULC change scenarios – to produce an overall analysis of GC (e.g., Pulido-Velazquez et al., 2014; Guo et al., 2015); some address cases of coastal aquifers (Benini et al., 2016; Gorelick and Zheng, 2015).

From a methodological point of view, the study of the impacts of potential GC on groundwater using an integrated and holistic climatic-agronomical-hydrological model that includes water quantity and quality continues to be a big challenge.

From an operational point of view, research aimed at solving these problems has focussed on sequential coupling of models. The assessment of these impacts requires models that can predict the evolution of the fresh water-salt water interface. In order to obtain an accurate representation of the physical process involved in seawater intrusion, flow and transport need to be coupled and solved simultaneously for each time step. Such an approach gives density-dependent solutions (e.g., Shammas and Thunvik, 2009; Doulgeris and Zissis, 2014) that take the salinization process into account. Simplified sharp interface models would provide a less accurate approach with a fewer number of parameters and lower computational requirements (e.g., Llopis-Albert and Pulido-Velazquez, 2014, 2015).

The objective of this work is to perform an integrated analysis of future potential GC scenarios (including CC, LULC change and SLR) and their hydrological impacts in a coastal aquifer, the Plana Oropesa-Torreblanca aquifer. We simultaneously

consider water quantity and quality in order to approach the salinization process. Section 2 describes the aquifer and the available data, while Section 3 presents the methodology. We propose a method to analyse and generate potential future GC scenarios involving different sources of uncertainty (Section 3.1). A modelling framework is defined to assess hydrological impacts on the coastal aquifer based on a sequential coupling of rainfall-recharge models, crop irrigations requirements and irrigation-return models, and a density-dependent solution that couples the resolution of flow and transport calculations for each time step (Section 3.2). Section 4 presents the results and their discussion. It includes an analysis of the sensitivity to potential future changes in rainfall recharge, LULC and sea level. We consider the limitations of this study and propose future research. Lastly, Section 5 presents the main conclusions of this research.

## 2. Materials: Description of the aquifer and the data available

### 2.1 Location and hydrogeology

The Plana Oropesa-Torreblanca is a shallow heterogeneous detrital aquifer that extends over approximately 75 km$^2$. It is oriented NE-SW, parallel to the Mediterranean coast along of 21 km, being its inland dimension in a range between 2.5 and 6 km. The ground surface over the aquifer comprises a gentle relief, steepening towards the surrounding limestone massifs (Figure 1).

Geographically, the Plana Oropesa-Torreblanca aquifer borders the Irta Mountain to the north (Cretaceous-Jurassic limestone), which is in hydraulic connection with the Plana. To its south lie the Oropesa Mountains (Cretaceous limestone). The western border (southern Maestrazgo) is formed by the Aptian and Gargasian limestone massif, which is in hydraulic connection with the Plana Oropesa-Torreblanca aquifer (except in the immediate vicinity of the Chinchilla and Estopet rivers where the impermeable Miocene base appears; Morell and Giménez, 1997; Renau-Pruñonosa et al., 2016).

The Plana Oropesa-Torreblanca is composed of Plioquaternary detrital materials comprising limestone pebbles, gravel and conglomerates derived from the adjacent mountain ranges, with abundant lenses of coarse sand, silt and clays. There are frequent lateral and vertical changes of facies and the overall distribution is irregular. The aquifer is overlain by more recent alluvial fans, colluviums, dunes and peatlands. Its geometry is lenticular – it is thinnest in the interior and thickest near the coast, exceeding 80 meters at the mouths of the Estopet and Chinchilla rivers. Several studies have demonstrated how the transmissivity of the aquifer varies over a wide range: from 5000 m$^2$/day to 100 m$^2$/day; the calculated effective porosity varies from 1% to 13% with the highest porosity nearest to the coastline. Inflows to the system consist of lateral groundwater transfers coming from adjacent aquifers, infiltration from precipitation and irrigation returns. Outflows comprise the pumped abstractions, together with groundwater discharges to sea and seeps/springs in the Prat de Cabanes wetland. Under natural conditions, groundwater flows NW-SE, perpendicular to the coastline (Morell and Giménez, 1997; Renau-Pruñonosa et al., 2016).

The carbonate basement receives the contributions coming from the bordering aquifer (Maestrazgo aquifer) and feeds the detrital aquifer. The direction of the groundwater flow is from the carbonate basement to the detrital aquifer.

The Prat de Cabanes is a wetland located in the center of the Plana. It extends approximately 9 km$^2$, parallel to the coastline with an elongated shape. It is separated from the sea by a coastal bar some 8 km long, 20 m wide and 3 m high, consisting of sorted pebbles. It is composed of brown and black silt and loam, with a recognized peat level some 3 to 4 meters thick, which is commercially exploited.

### 2.2 Historical LULC changes and climatic-hydrological data

In the 1960s and early 1970s the Oropesa-Torreblanca Plain was sparsely populated and land was dedicated mostly to non-irrigated cropping. From 1975-1995 there was a significant transformation from dry to irrigated lands, especially in the period 1985-1995. From 1995 to 2010, marked changes occurred in LULC, with a generalised overhaul of the irrigation

systems used and a conversion from agricultural to residential LULC, particularly along the coastal belt (Oropesa del Mar, Amplaries and Torre la Sal).

The following historical data for the period 1973-2010 were used to define the inputs of the coupled modelling framework described in Section 3:

1) Changes in LULC (Figure 2), obtained from both fieldworks undertaken in the area and from the European CORINE Land Cover database (Feranec et al., 2010). These data were used to estimate the irrigation returns, following the procedure described in Section 3.2.2.

A comparison of the land cover maps for 1990 and 2006 shows that the main change in LULC classes was an increase (+227%; from 182 ha in 1996 to 592 ha in 2006) of artificial surfaces (transport infrastructure, urban sprawl, tourism and recreation facilities), mainly at the expense of agricultural areas (-5.5 %; from 7420 ha in 1996 to 7015 ha in 2006). According to the Júcar River Basin Management Plan (CHJ, 2015), the percentage LULC for major crops are: citrus fruits 72.4%, vines 12.9%, outdoor vegetables 7.9% and others 6.8%. As for irrigation techniques, drip irrigation supplies 65.5% of the total irrigated area, flood irrigation provides 34% and spray irrigation provides 0.5%, with an overall irrigation efficiency of 60.1%.

2) Historical rainfall and temperature (T) for the Plana Oropesa-Torreblanca and Maestrazgo aquifers (See Figure 3) were taken from the Spain02 project dataset (Herrera et al., 2016). They were used to estimate rainfall-recharge (see Section 3.2.2.)

3) Historical evolution of total pumping in the Plana Oropesa-Torreblanca aquifer was deduced from historical data.

The transformation from dry to irrigated croplands led to an increase in pumped abstractions that extended over two decades (1975-1995, especially in the period 1985-1995), provoking a drop in groundwater level and seawater intrusion problems. From 1995 to 2010 there was a progressive reduction in pumping due to the abandonment of certain crops and irrigated areas.

A graphical representation of these series is included in Section 3.2.3 (density-dependent flow model).

Other hydrological information used to calibrate the models are:

1) An infiltration rate coefficient of 14% for the historical period, which was obtained from previous lysimiter readings from a neighbouring aquifer with similar characteristic (Plana de Castellón; Tuñon, 2000). It has been used to generate historical series of rainfall recharge by applying an infiltration coefficient directly to rainfall data, which is a simple approximation commonly applied (Kirn et al., 2016). The mean historical recharge (85 mm/year) obtained from this infiltration rate coefficient is quite similar to the mean (89 mm/year) estimated by other authors who applied an atmospheric chloride mass balance (Alcalá and Custodio, 2014).

2) Irrigation return coefficients for the crops in the area where taken from a previous study performed by Tuñon (2000). They have been used to assess return from irrigation demands.

3) Hydraulic heads and salinity in different observation wells.

The available observation network (21 points of hydraulic head and 31 points for salinity) yielded good distributed information on aquifer state from both, quantitative and qualitative points of view. The location of the observation wells and the evolution of the variables at some of these points are is described in Section 3.2.3 (density-dependent flow model). We have used the information available for the period 1973-2010.

## 3. Method. Application to the case study

The flowchart of the method has been represented in Figure 4. It summaries the steps that we propose to follow in order to perform an integrated assessment of potential GC scenarios. First of all, we propose and approach to generate future potential GC scenarios (section 3.1) considering, LULC, CC and SLR. Then, a modelling framework (Section 3.2) was defined to assess hydrological impacts on the coastal aquifer based on a density-dependent simulation whose inputs are

defined by sequential coupling of different models. Finally, it is used to propagate the generated potential future GC scenarios (Section 3.3).

## 3.1 Generation of future GC scenarios

They have been defined by combining the future LULC scenario approved in the General Town Plans in the area and the generated CC and SLR scenarios.

### 3.1.1 Future LULC change scenarios

The predicted future changes in LULC over the Plana Oropesa-Torreblanca are of greater magnitude than the historical ones and could drastically modify the rural and urban landscape. The already-approved tourist developments (the public urbanization work (PAI) for the Marina d'Or Golf in Oropesa and Cabanes, and the General Town Plan (PGOU) for Torreblanca) anticipate an increase in population of more than 130,000 inhabitants, as well as the disappearance of most of the agricultural activity in the area. These significant changes in LULC will produce significant impacts on water demands, and, therefore in pumping and recharge and so to the hydrodynamics of the aquifer. In contrast, there are no significant changes to LULC anticipated in the area belonging to the municipality of Alcalà de Xivert, also situated on the Plana.

The General Town Plan for Torreblanca (PGOU Torreblanca, 2009) approves the conversion of 70% of the municipality's area included in the plan, which is currently used for citrus agriculture, into land classified as buildable residential or industrial. Even along the coast, to the north of Prat de Cabanes, is the projected urbanization – included in the so-called 'Integrated Activity Plan' (PAI) – of Doña Blanca Golf (Figure 5).

The municipalities of Cabanes and Oropesa in the southern inland part of the Plana de Oropesa-Torreblanca have approved PAIs for the Marina d´Or Golf, which will include three golf courses, private residential developments, hotel complexes and associated garden areas. Once all these planned constructions are completed, they will cover some 16 km$^2$ (Figure 5). Two of the conditions that the Valencian Government imposed before it approved the PAI Marina D´Or Golf are that the whole of the area that requires irrigation (both golf course and garden areas) must use recycled residential water from the wastewater treatment works; and that all water destined for urban supply must be sourced from a desalination plant. Thus, as the various individual projects are built, the groundwater abstractions in the area are falling, until they will cease completely at the end of the PAI development period. Over the neighbouring Maestrazgo aquifer, we will assume that there are no changes in LULC.

### 3.1.2 Generation of potential future climate scenarios for the system

We propose a method to generate consistent potential future climate scenarios for a short-term horizon (2011-2035) from the historical (1973-2010) data (see section 2.2) and the climate models simulations performed in the frame of the CORDEX EU project (2013). It requires an analysis of the results obtained by applying different downscaling techniques. A multi-criteria analysis of some statistic of these series was performed to identify the best simulations of the historical data. Different ensembles hypothesis have been adopted to define more representative potential future climate scenarios to be employed in the groundwater impacts study. Our main target is to provide an estimate of the most representative plausible future climate scenarios. For this reason we propose to generate and propagate 4 plausible representative climate scenarios defined by ensemble of different climate models, which provide a better approach of future climate scenarios than taking directly a scenarios defined by a single model. The 'ensembles' coalesce and consolidate the results of individual climate projections, thus allowing for more robust climate projections that are more representative than those based on a single model (Spanish Meteorological Agency, AEMET, 2009). In this paper we do not intend to perform a detailed analysis of hydrological

uncertainty. In this case, in order to assess the uncertainty on hydrological impacts it would be more appropriate to obtain results from each individual climate model.

- **Climate model simulation data. Control and future scenarios.**

In this work we have focused on the information available for the most pessimistic emission scenario (RCP8.5) We analyse the information coming from EU CORDEX project (2013), where we find nine climate-change scenarios (see Table 1) defined with the simulations (control and future series) obtained with five Regional Climate Models (CCLM4-8-17, RCA4, HIRHAM5, RACMO22E and WRF331F) nested with some GCMs (4 GCM were available).

We have obtained representative lumped series of these simulations for our system, by weighting the values in each CORDEX cell according to its surface in the domain. Figure 6 shows a significant bias between the historical data and the control simulation that will force us to apply a correction technique to generate future scenarios.

We also observe important differences between the statistic of future (2011-2035) and control (1976-2000) series (rainfall and T) for each RCM (Figure 7).

- **Application of different downscaling techniques (bias correction and delta change techniques)**

In accordance with the hypotheses assumed to define future climate series in a water resource system (starting from the climate model simulations) we can consider two different kinds of downscaling techniques: bias correction techniques and delta change techniques (Räisänen and Räty, 2012). In the present study we apply both conceptual approaches (bias correction and delta change techniques).

The bias correction techniques are based on the analysis of the statistical difference between the climatic variables in the historical data and the control simulations produced by the climate models for the same period. They aim to define a transformation function to correct the control series to obtain a better approximation of the historical statistic. They assume that in the future the bias between model and data will be the same as observed in the historical period (e.g., Watanabe et al., 2012; Haerter et al., 2011). The delta change approaches assume that the model can obtain a good approximation of the relative changes in climate variables statistics, but do not provide a good prediction of the absolute values. Accordingly, they try to characterize the 'delta change' produced in the main statistics of the climatic variables by analysing the relative difference between the future and control scenarios simulations. The future series will be obtained by perturbation of the historical series in accordance with the estimated 'delta change' (e.g., Pulido-Velazquez et al., 2014, 2011; Räisänen and Räty, 2012).

When applying both correction techniques (bias and delta change) the spatial resolution of the historical data available for our systems is usually more detailed than those adopted by the climate model and, therefore, these transformations indirectly produce solutions with higher spatial resolution than the RCM one. Therefore, they are commonly known as downscaling transformations. We have applied two downscaling techniques (correction of first and second order moments) for both conceptual approaches (bias correction and delta change techniques). The correction of the first and second moments for the delta change approach was defined at monthly scale as described in Pulido-Velazquez et al. (2014). The bias correction case was also defined in an analogous way. The difference is that the perturbation is defined by modifying the statistics (first and second moments) of the control series to approximate the historical ones. It will be applied assuming to correct the future assuming that will be invariant. As example, the first, second and third moments of the series obtained with both approaches for one of the RCM (RCA4 linked to CNRM-CM5) are represented in Figure 8.

- **Multi-criteria analysis of the main statistic**

Two multi-objective analyses are proposed: one related with the bias approaches and another with the delta ones.

In the delta change approaches we applied the multi-criteria analysis proposed by Pulido-Velazquez et al.. (2014). It intends to identify the models that provided the best approximations to the main historical statistics (mean, standard deviation and asymmetry coefficients) based on the analysis of their control simulation of the historical period. The dominated solution or 'inferior' models approaching the historical statistic were identified and eliminated. A model is eliminated (see Table 2) if any other model's prediction provides better approximation to the cited statistics.

In this study we also propose a multiple-criteria analysis for the bias correction approaches. This allows us to identify the best combination of model and bias correction techniques (see Table 3) to approximate the main statistics of the historical series. Since most of the combinations of model and bias correction technique provide very good approximations to the first moment, a relative error threshold was defined to consider a corrected control to approach better an statistic when significant differences (higher than the threshold) are obtained.

- **Ensembles of predictions to define more representative future climate scenarios**

We considered four options to define representative future scenarios by applying different ensembles of corrected projections. Two ensemble scenarios ($E_i$) were generated by a linear combination of all the future series generated by delta change ($E_1$) or bias correction ($E_2$). Two other options were defined by combining only the non-eliminated models (E3, for the delta change approach) or combinations of models and correction techniques (E4, for the bias correction techniques). They do not consider the eliminated options because they provide inferior approximations to the historical series that make us mistrust their predictions.

All ensemble predictions show very similar increase in mean T (see Figure 9). The standard deviation estimated with the delta change approaches are quite similar to the historical, but both ensembles defined by applying bias correction show smaller standard deviations.

A reduction in future mean rainfall is predicted by all the ensembles for every month except September and October, when relative increases in rainfall are predicted. Since these are the months with highest historical rainfall, the overall effect is a higher total annual rainfall compared to the historical. During September and October in this Mediterranean area we have important storms related with the phenomenon known as the "cold drop" (Roth, 2003). It is related with the higher total rainfall observed in these months. In accordance with the obtained potential scenarios, in the future we would have an increment of these extreme rainfall events.

As we also observed for the T, the standard deviation of the future rainfall predicted with the delta change approaches are quite similar to the historical one, but both ensembles defined by applying bias correction show significant reductions.

**3.1.3 Sea Level Rise (SLR) scenarios**

Based on the European Environment Agency analysis (EAA 2014) of historical and potential future SLRs, we propose a SLR scenario to study the sensitivity of our GC simulations to a potential SLR. The EAA report states that, over the last two decades, satellite measurements have indicated a mean rate of SLR of more than 3.2 mm/year. If we assume that this rate remains constant then, by the end of a future horizon of 25 years, we would have a rise in sea level of 0.08 m. On the other hand, model simulations for the RCP8.5 emission scenario show a rise in sea level for 2081-2100 in the range 0.45-0.81 m. If we assume a constant rate of SLR from now until 2100, the sea level would rise a maximum of 0.19 m (more than double the observed rate over the last two decades). This value (a rise of 0.19 m by the end of the future horizon) was used to define a very pessimistic scenario of maximum SLR. A linear SLR was considered during the future horizon (2011-2035).

**3.2 Definition of a coupled modelling framework**

In order to assess quantity and quality impacts on groundwater systems we needed to calibrate a density-dependent model that simulates flow and transport within the porous aquifer medium. We propose a sequential coupling of three 'auxiliary models' (rainfall-recharge models and crop irrigation requirements and irrigation returns models) with this density-dependent model, in which the outputs of the auxiliary models are used as inputs of the groundwater model (see Figure 4).

The models were calibrated with the available historical data (1981-2010). Historical data for the period 1973-1981 were used to validate them. These models were then used to simulate the impacts of future LULC and CC scenarios.

### 3.2.1 Rainfall-recharge models

Based on the historical climate (rainfall and T) and recharge series for the period 1973-2010 described in sections 2.2, we applied the simple empirical rainfall-recharge proposed by Pulido-Velazquez et al., 2017  to generate yearly aquifer recharge

series. The model assumes that rainfall and T are the most important climatic variables determining potential aquifer recharge, and the variability of both of them will determine the impacts of future potential climatic scenarios on the recharge It intends to define a correction function for the perturbation of the historical series defined as the difference in rainfall and evapotranspiration (ETR) (difference hereafter referred to as PE series), modifying its mean and standard deviation to make them equal to the statistic of the historical aquifer recharge previously deduced from the lysimeter measurements (see section

2.2). The calibrated function is employed to obtain the precipitation recharge series (PR) also for the future period. The steps required are (Pulido-Velazquez et al., 2017):

1) Estimation of the change in mean and standard deviation of PE and PR series for the historical period

$$\Delta\mu = \frac{\mu(PR)-\mu(PE)}{\mu(PR)} \text{ and } \Delta\sigma = \frac{\sigma(PR)-\sigma(PE)}{\sigma(PR)} \tag{1}$$

2) Standardization of the PE series (historical and future)

$$PEn_i = \frac{PE_i-\overline{PE}}{\sigma_{PE}} \tag{2}$$

3) Generation of PR series from PE series

The future R series will be obtained from the future PR series assuming that the bias ($\Delta\mu$ and $\Delta\sigma$) remain invariant in the future.

$$PR_i = \sigma_c \cdot PEn_i + \mu_c \tag{3}$$

Where $\mu_c = \mu(PE) \cdot (1 + \Delta\mu)$ and $\sigma_c = \sigma(PE) \cdot (1 + \Delta\sigma)$ $\tag{4}$

Taking the positive relationship of T and E into account (Arora, 2002; Gerrits et al., 2009), changes in T will determine the available non-evaporative fraction of rainfall available for aquifer recharge. Different non-global empirical models could be

applied to assess the historical E from T series (e.g., Turc, 1954, 1961; Coutagne, 1954; Budyko, 1974; amongst others) as described in Arora (2002), Gerrits et al. (2009), and España et al. (2013). In this study, we applied Turc's model (1954, 1961), in which the results depend on mean annual T and solar irradiation over the latitude.

In order to approach the impact of seasonal variability the annual rainfall recharge values obtained using the simplified model were distributed between the 12 months maintaining the pattern of the historical rainfall recharge series. We assume

that recharge from a rainfall event will reach the aquifer in less than one month, so working with stress periods of one month means there is no delay between the rainfall and the aquifer recharge.

Two simplified rainfall recharge models were developed, one for the Plana de Oropesa-Torreblanca aquifer and the other for the Maestrazgo aquifer.

The Maestrazgo aquifer recharge model is used to assess inflows to the carbonate basement, that produces lateral inflows (LI) to the Plana Oropesa-Torreblanca aquifer under various potential GC scenarios. The Maestrazgo aquifer has an important storage capacity which is almost in natural regime and does not present significant changes in hydraulic head. The Maestrazgo rainfall recharge model is employed to assess future recharge being the LI to the carbonate basement obtained by assuming a constant ratio between Maestrazgo rainfall recharge and these LI.

### 3.2.2 Modelling crop irrigation demands and irrigation returns

The LULC information was used to estimate agricultural water requirements following a procedure to compute crop water requirements based on the FAO Irrigation and Drainage Paper (Allen et al., 1998). This approximation was applied in previous CC impact research studies (e.g., Escriva-Bou et al., 2016). The irrigation values added to the rainfall constitute the total inflows coming from the surface system. A Turc model (1954) was employed to estimate the total ETR in the area considering not only rainfall but also irrigation water. The difference between the total ETR and the ETR for the rainfall allows us to determine the ETR related to irrigation, taking into account the climate conditions (rainfall and T). The estimated irrigation demands have been also employed to assess pumping taking into account information about the origin of the water that supplies each demand. The irrigation demands are multiplied by the irrigation return coefficients obtained for the crops in this area in previous studies (Tuñon, 2000) to assess recharge from irrigation.

### 3.2.3 Density dependent flow model (flow and transport)

Based on the hydrological description performed in Section 2.1 a conceptual model was defined to approach the aquifer as an unconfined heterogeneous detritic aquifer. The inflows to the aquifer include rainfall recharge, Lateral Groundwater Inflows from the bordering aquifers (LGI) and irrigation returns. The outflows include natural springs that feed the wetland, pumping wells and outflows to the sea. The historical evolution of the inflows and outflows has been represented in Figure 10.

The 3D finite-difference numerical code SEAWAT (Guo and Langevin, 2002) was used to solve the coupled partial differential equations for variable-density flow and transport. It combines MODFLOW (McDonald and Harbough, 1988) and MT3DMS (Zheng and Wang, 1999) into a single code that conserves fluid mass, rather than fluid volume, and uses equivalent freshwater head as the primary dependent variable.

The domain has been divided into 32 columns and 90 rows in the horizontal plane, with a cell size of 250x250 m. Therefore, the groundwater flow and transport domain extends over a size of 8000x22500 m. We adopted a vertical discretization consist of 11 layers in order to avoid a high computational cost (e.g. Sreekanth and Datta, 2010) while maintaining reasonable levels of numerical dispersion and representing complex flow patterns near areas of high concentration gradients (Guo and Langevin, 2002) . These 11 layers, were defined as confined/unconfined layers where the transmissivity varies. On the one hand, for the flow boundary conditions a prescribed head of 0 m has been assigned to all cells belonging to the sea frontier of the model. For the transport boundary conditions a prescribed concentration of 35 g/l has been assigned to all cells belonging to the sea frontier of the model.

We assumed a density of fresh water of 1000 kg/m3 and 1025 kg/m3 for the seawater of; with 0.7143 as the slope of the linear equation of state that relates fluid density to solute concentration.

The groundwater model covers the Pli-Quaternay and Pre-Quaternary formations. The spatial heterogeneity is tackled using the concept of multiple statistical populations (Llopis-Albert and Capilla, 2010), in which the rock matrix and each fracture is represented as independent statistical population. The random function for each structure (i.e., the aquifer matrix and fractures) is modeled based on a geostatistical analysis conditionated to its own statistical distribution (i.e., hydraulic

conductivity data as well as geological information and expert judgment). The random function is supposed to be as MultiGaussian for the rock matrix, while the fractures are considered as non-MultiGaussian. In this way, the rock matrix is generated by sequential Gaussian simulation using the code GCOSIM3D (Gómez-Hernández and Srivastava, 1990), while the fractures are generated by sequential indicator simulation using the code ISIM3D (Gómez-Hernández and Journel, 1993).

The latter code makes use of local conditional cumulative density functions (ccdfs) defined by conductivity measurements and the corresponding indicator variograms. Therefore, the spatial heterogeneity is modeled as an equivalent porous media (e.g., Llopis-Albert and Capilla, 2010). On the one hand, the hydraulic conductivity data for fractures presents high values, greater than 1000 m/d. This allows the reproduction of strings of extreme values of hydraulic conductivity that often take place in nature and can be crucial in order to obtain realistic and safe estimations of mass transport predictions. That is, it allows reproducing preferential flow channels in strongly heterogeneous aquifers or fractured formations. On the other hand, the hydraulic conductivity data for the aquifer matrix cover a wide range of values, i.e., from 5 to 200 m/d. In addition, for each cell we have defined a vertical hydraulic conductivity equal to a tenth of the horizontal hydraulic conductivity. The position of fractures is deterministically incorporated in the model based on geological information and expert judgment, thus allowing to classify the cell models. Those cells of the model intersected by a fracture are assigned conductivities according to the intersecting fracture, and those that are not are considered as cells belonging to the rock matrix.

The model was calibrated with the available historical data in the period (1981-2010), while the data in the period 1973-1981 was used to validate it. This was carried out by applying a trial and error procedure simultaneously considering quantity and quality. We have not used an inverse model (eg. Llopis-Albert et al., 2016) because its computational cost would be important in order to deal with such quantity of parameters (hydraulic conductivity, storativity, porosity, dispersion coefficients …) and variables (hydraulic head and salinity) within a seawat model over a long period of time (from 1973 to 2010), so that we have decided to apply a trial and error procedure.

The calibrated model parameters encompass different zones of horizontal hydraulic conductivities ranging from 5 to 200 m/d, while the vertical hydraulic conductivities are between 0.5 to 20 m/d. There are also different zones of specific storage values, which range from 10-5 to 5·10-4 1/m; specific yield, with values from 0.01 to 0.15; effective porosity values from 0.01 to 0.13; and dispersion coefficients from 50 to 100 m. We have replaced a highly heterogeneous porous media with an upscaled "equivalent" homogeneous porous media to represent the hydrogeological parameters since the cell size of the discretization is 250x250 m (e.g., Llopis-Albert and Capilla, 2010). Then, we have used a value of the effective porosity based on available data, which were subsequently calibrated and upscaled using expert judgment. The results of the calibration process prove the worth of this approach.

Reasonably good results in terms of goodness of fit to the historical hydraulic head and salinity were obtained, as shown in Figure 11 for various head and salinity observation wells.

The root mean square value (RMS) of the departures between observed and simulated values for both, piezometric heads and salt concentrations, is presented for the whole domain and temporal discretization due to the large number of boreholes (21 boreholes for piezometric heads and 31 for salinity). These values are $\eta_h \eta_h = 0.7$ m; $\eta_c \eta_c = 391.8$ mg/l.

Note that this $\eta_h \eta_h = 0.7$ m could seems to be a little high but we should take into account that we have observation wells where the historical hydraulic head measurement fluctuates sometimes more than 6 m during the same month (see for example observation well 6). Note that the high value for $\eta_c \eta_c = 391.8$ mg/l could be also explained by the scale of the salt concentration, which range from 0 to 35000 mg/l, and the measurement fluctuations, which in some wells is even higher than 4000 mg/l during some months.

Finally, the calibrated model is used to propagate impacts of future LULC and CC scenarios.

**3.3 Hydrological impacts: propagation of future climatic scenarios**

Analysis of GC impacts will require a sequential simulation of the LULC change scenarios for the different climate scenarios ($E_i$) in the auxiliary models (rainfall-recharge models and the crop irrigation requirement and irrigation returns models) in order to define the inputs for the groundwater density-dependent model.

In order to assess the potential impacts of the future LULC scenario (LULC scenario) and different GC (CC and LULC change scenarios) we have simulated the following scenarios using the density-dependent flow model:

1) Baseline (BL) scenario: No LULC change and no CC. We simulate a future scenario for the horizon 2011-2035 assuming that from 2011 we would have the same LULC that we observed in 2010. We also assume that the hydrological characteristic does not change and we have simulated assuming the rainfall recharge and the LI from the neighbour aquifer are equal to those estimated in the last 5 years of the historical periods (2006-2010). In this period of 5 years (2006-2010) there was no significant change in LULC and so this period could be adopted as being representative of the mean recent climatic-hydrological conditions. This scenario was defined in order to compare against the others to analyse the sensitivity to GC.

2) LULC scenario: It considers the described future LULC scenario and assumes that there is not CC.

3) Global change scenarios ($GC_i$) assuming constant sea level: We consider four GC scenarios that simultaneously consider the potential impacts of the described future LULC scenario under the four generated CC scenarios (Eisee Section 3.1.2.4). The comparison of these scenarios with the BL provides information about the GC impacts.

4) SLR scenarios: We have also simulated the four GC scenarios assuming that the sea level grows in a linear way 0.19 m during the future horizon (2011-235).

The results obtained are summarized and discussed in next section.

## 4. Results and discussion

The proposed approach has similarities with the one described by Pulido-Velazquez et al. (2015), in which an integrated analysis of GC is performed including CC and LULC changes. The most important differences are related with the fact that a coastal aquifer is studied and we consider quality issues simulating with a variable density flow and transport model. The model was calibrated with the historical data. The approach allows to propagate impacts of different CC and LULC change scenarios in terms of global flow balance, as well as a distributed approximation of the hydraulic head and salinity.

The historical simulation in accordance with the data shows that, in most of the aquifer area (more than 80 %), the salinity is above the natural background of the aquifer (1100 mg/l), and, therefore, it is in general affected by intrusion.

The GC scenarios have been analyzed in terms of impacts on global flow balance components and salinity at specific points.

The four GC scenarios show higher variability in all the inflows and outflows in the aquifer (See GC1 in Figure 12). If we focus on the components of the global balance, in global terms, this analyses does not show an increment of the intrusion, and even, it would be slightly reduced with respect to the Baseline scenario due to the outflows to the sea will not decrease (see Figure 13). These results show a lower sensitivity to the LULC than to the GC scenarios in which the impacts also include Climate Change.

Figure 14 shows the evolution in terms of salinity at 4 specific observation points roughly equispaced. They were selected to cover the extension of the aquifer from north to south (starting with the more northerly and moving towards the south we have observation wells 33, 12, 39 and 21 respectively).

From the results in terms of salinity at specific observation points (Figure 14), we can observe the heterogeneity of the impacts of the LULC scenario and CC scenarios. The area around the observation point 33 is not affected by LULC changes and we only observe sensitivity to CC scenarios that produce a higher variability in the salinity evolution, but the mean trend of the concentration does not change. Around the observation point 12, in Torreblanca area, the LULC change scenario would produce a reduction in the recharge whose impacts can be observed as an increment in the salinity during the first decade of the future horizon. Nevertheless, in the last 10 years the reduction in pumping produced by the successive

transformations of irrigated areas to residential land would reduce significantly the salinity. In this Torreblanca area CC scenarios would impact the salt concentration significantly during the first years due to the increment in pumping requirements produced by the higher water irrigation requirements obtained for these CC scenarios. Even considering the impacts of CC scenarios, during the last 10 year of the horizon it starts to recover due to the reduction in pumping produced by the new transformations of irrigated areas to residential land defined in the LULC scenario, being the concentration at the end of the horizon (2035) even under the values obtained for the BL scenario. In the observation point 39, located further away from the coast, the reduction in pumping due to LULC change would reduce the salinity during most of the year of the horizon. Nevertheless the reduction in recharge makes in some years the salinity to get close to the one obtained without LULC changes (BL scenario), being very similar at the end of the period. Again CC scenarios would increase the variability of the salinity in the simulated period.

We can also identify in the southern part areas where the situation will clearly improve throughout the future horizon contemplated with the proposed scenarios. For example, at observation point 21, in Oropesa area, the salinity would be reduced with the contemplated scenarios, which would be mainly related with the reduction of pumping in this area due to the transformation of irrigated areas to residential land defined in the LULC scenario.

As commented above the expected results considering GC impacts are likely to be too pessimistic or optimistic, depending on the location. These results can be useful for the authorities in charge of implementing management policies in the Plana de Oropesa Torrablanca. We can use this coupled modeling framework to assess potential effect of adaptation measures to GC by modifying the inputs of the models. Participatory processes including the relevant stakeholders might be essential in the definition of scenarios and successful adaptation measures (Pulido-Velazquez et al., 2015). This modeling framework could be useful in the search for consensus ("shared vision" models) between different stakeholders.

Lastly, we analyzed the sensitivity of the GC scenarios to a SLR scenario (Figure 15). A rise of 0.19 m at the end of the future horizon was used to define a very pessimistic scenario of maximum SLR (see Section 3.1.3). A linear SLR was considered during the future horizon (2011-2035). Figure 15 shows the results obtained in terms of hydraulic head and salinity at a number of observation points. It shows that the sensitivity of hydraulic head is very low. The sensitivity of the salinity, although not very significant, is higher than that observed for hydraulic head. The low sensitivity of the results should be due to the maximum value of SLR considered, 0.19m in 2035, is quite low with respect to the level fluctuations experienced in most of the observation wells (see Figure 15). For this reason, the sensitivity of the flow and transport solutions are low. We find in the literature other examples in which the sensitivity of seawater intrusion to the SLR would be low. Chan et al. (2011) obtained this conclusion in a synthetic confined coastal aquifer in which recharge in unchanged; Rasmussen et al. (2013) obtained the same conclusion for an inland coastal aquifer with minor SLRs. Nevertheless other authors, as Werner and Simmons (2009) showed that in unconfined aquifers the influence of the inland boundary condition can be significant to its sensitivity to SLR.

The methodology proposed in this paper to perform an integrated analysis of future potential GC scenarios (considering, LULC, CC and SLR) and their hydrological impacts is general. It can be used to assess the potential status of any coastal aquifer in terms of flow balance components, hydraulic head, and salinity. It includes the definition of future CC scenarios by using equi-feasible and non-equifeasible ensemble of projections based on the results of a multi-criteria analysis of the series generated from several Regional Climatic Models with different downscaling approaches. A modelling framework was proposed to assess hydrological impacts of future climatic scenarios on the coastal aquifer based on a density-dependent simulation whose inputs are defined by sequential coupling of rainfall-recharge models, crop irrigations requirements and irrigation returns models (a chain of models). This chain of models, calibrated using the available historical data, allow testing the conceptual approximation of the aquifer behavior and the propagation of the generated potential future scenarios.

The application of the proposed method allows to improve our knowledge about the case study and so comprise a tool to support decisions about sustainable adaptation management strategies. In the definition of scenarios and plausible adaptation

measures to be analyzed participatory processes including the relevant stakeholders might be essential (Pulido-Velazquez et al., 2015). This modeling framework could be useful in the search for consensus ("shared vision" models) between different stakeholders in the definition of the cited scenarios. The limitations and potential future extensions of this work, for example in terms of analyses of uncertainties, are discussed in the next subsection (4.1).

## 4. 1 Hypotheses and limitations. Usefulness of the results.

From the methodological point of view, the proposed approach is general, ambitious and valid scientifically. Nevertheless there are some assumptions and limitations in the application performed that we wanted to highlight and summarize in this section in order to clarify the accuracy and utility of the results obtained. We have grouped them in three categories:

(1) Generation of future potential climatic scenarios

- The research is focused on the analysis of future short-term-horizon (2011-2035). Other future horizons, such as mid-term and long-term scenarios, in which there should be even higher uncertainties about the impacts of CC for being further away in time, are not considered.

- Only the most severe IPCC scenario (RCP. 8.5) was analyzed to assess the most pessimistic impacts in the future short-term scenario.

- Potential plausible future climate scenarios are defined by combining information coming from different Regional Climatic Models (RCMs) and General Circulation models (GCMs).

- Two downscaling approaches (correction of first and second order moments) under two different hypotheses (bias correction and delta change techniques) were applied to generate future climatic series in accordance with RCM simulations. Note that, depending on the problem and the target solution, several downscaling techniques of varying complexity and accuracy (correction of first- and second-order moments, regression approach, quantile mapping, etc.) can be applied by assuming different conceptual approaches, such as bias correction and delta change techniques (Räisänen and Räty, 2012).

- Two different hypotheses, equifeasible members or non-equifeasible members, were applied to define ensembles of the obtained future series for each RCM. They may help to achieve more representative future potential climate scenarios for assessing impacts on aquifer recharge.

(2) Hydrological propagation of the climatic impact to aquifer recharge

- A simple empirical precipitation-recharge model has been adopted; whose inputs are rainfall and E. We have not tested other hydrological models, for example, based on a more physically based or detailed representation of the processes involved in the hydrological balance and the geological structures. We assume that rainfall and T are the variables determining NAR, and their spatiotemporal variability determines the impacts of future potential climatic scenarios on renewable groundwater resources. We do not consider the change changes in other variables affecting recharge, such as soil properties, vegetation patterns and land-use. They are considered steady, despite there being expected to change according to global climate driving forces and new human actions on a local scale that will be induced by human adaptation to climate and water resource availability (Martínez-Valderrama et al., 2016).

- We assume that the climatic fields (rainfall and T) taken from the Spain02 project (Herrera et al., 2016) are good enough to approximate the historical climate.

- The Turc's model (1954, 1961) was applied to estimate E. Its results depend on mean annual T and rainfall. Different non-global empirical models could be applied to assess the historical E from T and rainfall time series.

(3) Hydrological impacts on aquifer status: Seawater intrusion simulation

In this study we decided to employ a variable-density model instead of a sharp-interface solution, which have been extensively used to define management models because of its simplicity in terms of required parameters and computational burden (e.g., Mantoglou et al., 2004). This is because they better describe the dynamic of real complex coastal aquifers

despite the limitations in the available data and the course discretization used for the Plana Oropesa-Torreblanca aquifer. Better adjustments between observed and modeled hydraulic head and salinity would provide greater confidence in the model predictions. Nevertheless, although due to there are some differences between observed and simulated data the uncertainty of the prediction coming from the model grows and we have to be cautious with the conclusions obtained, the fit is good enough to capture the general trend of the hydraulic head and salinity variables within a quite long calibration period (37 years, from 1973 to 2010), and therefore, to assess general impacts of climate and LULC changes. Note that it is difficult to improve the calibration of the proposed approach in this work for Plana de Oropesa-Torreblanca due to the following facts:

- The hydrogeological complexity of the aquifer makes it difficult to define a better approach taking into account its spatial heterogeneity, with fractured formations and preferential flow channels existing in the aquifer (Morell and Giménez, 1997). The spatial heterogeneity is handled by means of sequential indicator simulation using the computer code ISIM3D (Gómez-Hernández and Srivastava, 1990).

- The quality of some observation data is poor. This problem is accentuated when considering the long simulated time period with historical data, which spans from 1973 to 2010. In this way, for a certain observation borehole there are data close in time with measurements quite disparate, which cannot be explained by any physical phenomenon. A statistical processing of data with expert judgment would be advisable to dismiss the wrong data. We have opted for using all available data (as a transparency measurement and in order to ease the reproduction of this exercise by other researchers).

- The lack of reliable estimates of dispersion coefficients (Naji et al., 1999) may prevent a proper adjustment for the Plana Oropesa-Torreblanca aquifer.

- A better fit might be achieved using a more refined spatial discretization, which would allow modeling the preferential flow channels existing in the aquifer (Morell and Giménez, 1997). However, the high computational burden when solving the variable-density flow and transport equations with SEAWAT prevents the use of a fine grid (e.g. Sreekanth and Datta, 2010).

- As a further research, we are intended to couple the seawater intrusion model with a management simulation–optimization model for control and remediation that would further prevent the use of a fine grid.

    (4) Analysis of uncertainty in impacts

Note that, the information about seawater intrusion status provided by the analyses of flow balance is quite limited. We have not used indices based on the salinity distribution to assess the global intrusion status (see Baena-Ruiz et al., 2018). Finally, this paper we do not intend to perform a detailed analysis of uncertainty in impacts, which could be deeply analyzed as a further research. In order to assess the uncertainty on hydrological impacts it would be more appropriate to assess results from each individual climate model. Note that it would require to deal with different sources of uncertainty (Matott et al., 2009). The complexity is even greater for the presented methodology, since it entails the coupling of several numerical codes and a large amount of data and a long simulation time period.

**5. Conclusions**

We have proposed a general method to perform an integrated analysis of the potential impacts of future CC, LULC change and SLR in a coastal aquifer. It is assessed in terms of flow balance components, hydraulic head, and salinity distribution. It includes the definition of future CC scenarios by using equi-feasible and non-equifeasible ensemble of projections based on the results of a multi-criteria analysis of the series generated from several Regional Climatic Models with different downscaling approaches. A modelling framework, defined with a chain of models was proposed to assess hydrological impacts of these future climatic scenarios.

It has been applied in the Plana Oropesa-Torreblanca aquifer assuming some hypotheses or simplifications. Representative future CC scenarios are generated by applying the proposed method A future LULC scenario was defined in accordance with the plan approved by the local government (PGOU, 2009). Four GC scenarios were defined by combining the LULC scenario and the CC scenarios. These GC scenarios have been propagated to assess hydrological impact by simulating them within a coupled modelling framework based on density-dependent model whose inputs are defined by a sequential coupling of different models (rainfall-recharge models, crop irrigations requirements and irrigation return models). The historical status shows that the majority of the aquifer is affected by seawater intrusion. The analysis of the future flow balance components does not show an increment of the global intrusion. These results show a lower sensitivity to the LULC than to the GC scenarios in which the impacts also include Climate Change. These global scenarios' simulations show a significant increase (respect to a BL scenario with no GC) in the variability of the flow budget components and in the salinity. The impacts on the aquifer salinity will be heterogeneous. We observed specific areas where the situation gets worse and other, where it will clearly improve The proposed analysis is valuable to improve our knowledge about the aquifer and so comprise a tool to support decisions about sustainable adaptation management strategies. From an operational point of view, this modeling framework could be also useful in the search for consensus ("shared vision" models) between different stakeholders involved in the management of the aquifer.

**Acknowledgments:** This research work has been partially supported by the GESINHIMPADAPT project (CGL2013-48424-C2-2-R) with Spanish MINECO funds.

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

**Figure 1: Location map area and cross sections of the study.**

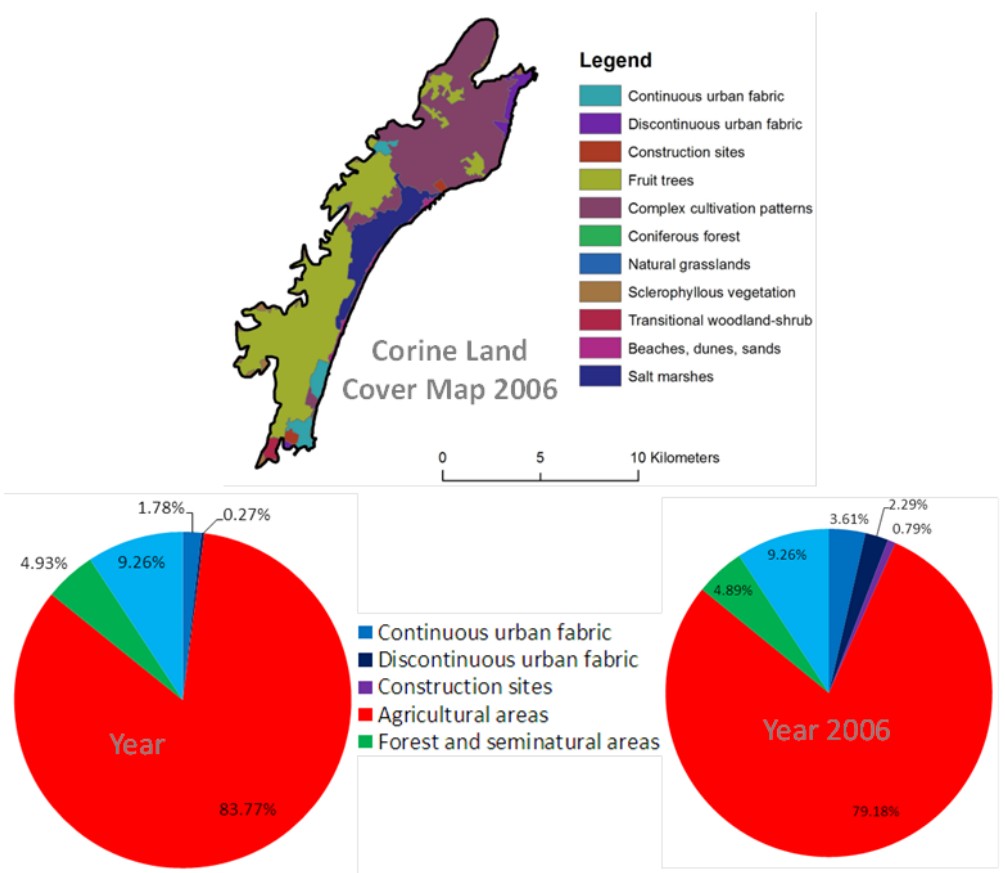

**Figure 2: Land use change (comparing the CORINE Land Cover databases for 1990 and 2006) over the Plana de Oropesa-Torreblanca and Maestrazgo aquifers.**

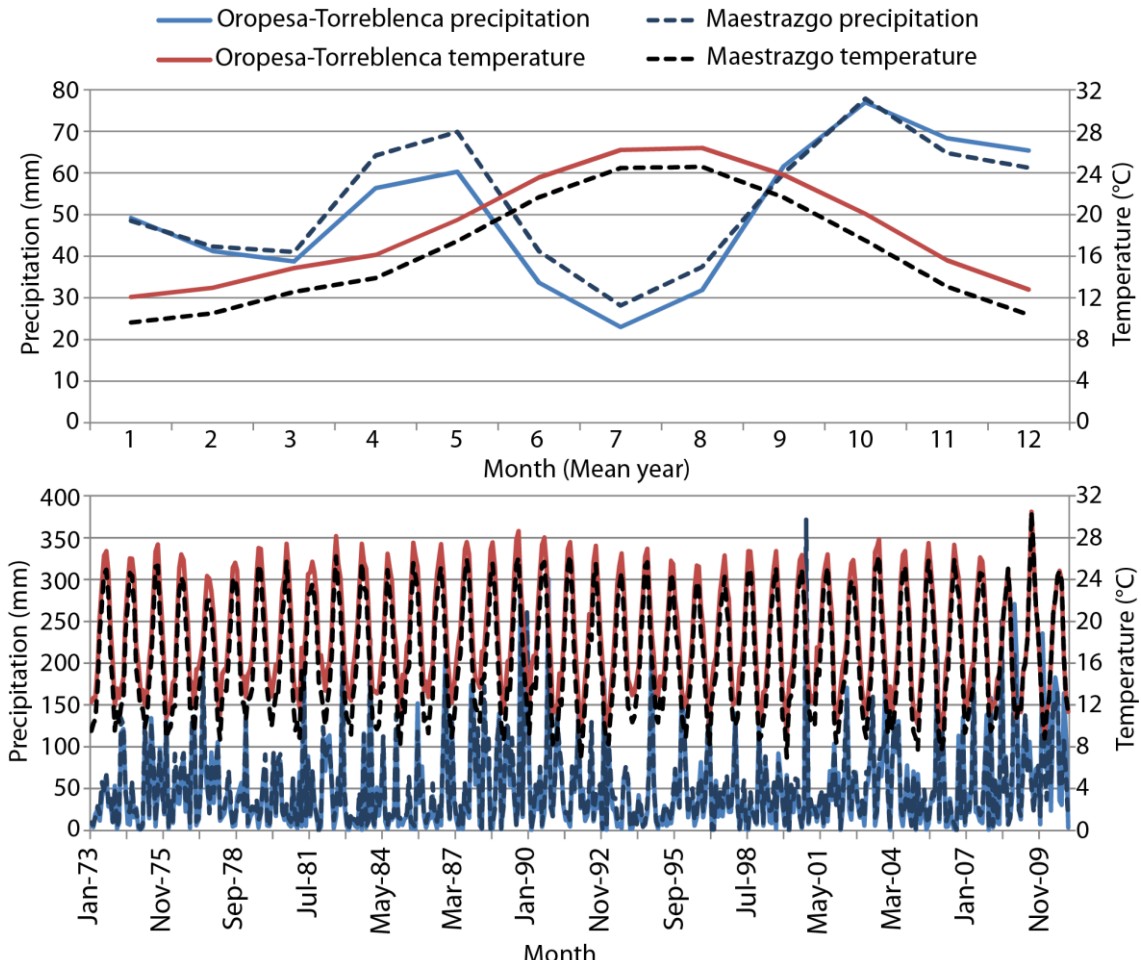

Figure 3: Historical rainfall and temperature in the Plana de Oropesa-Torreblanca and Maestrazgo aquifers.

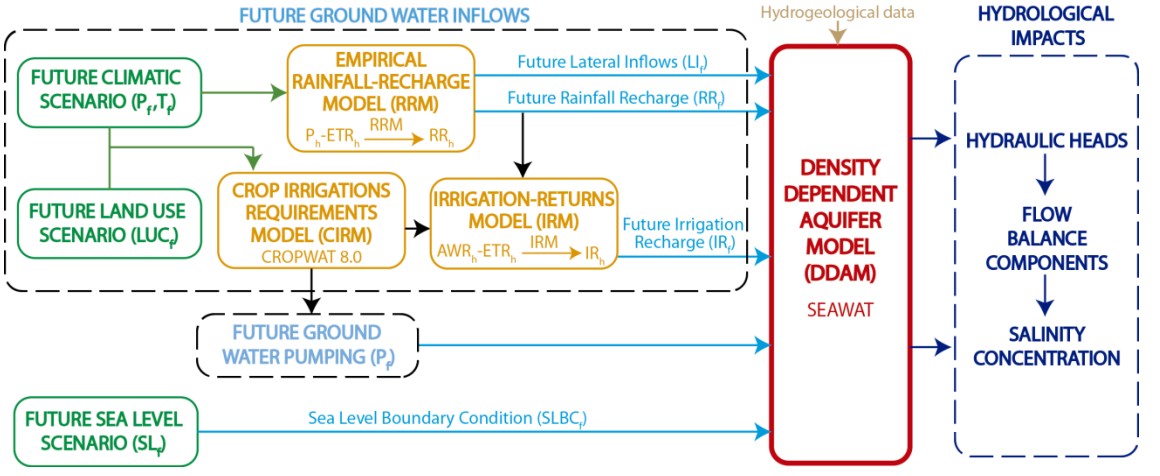

Figure 4: Flowchart of the modelling framework.

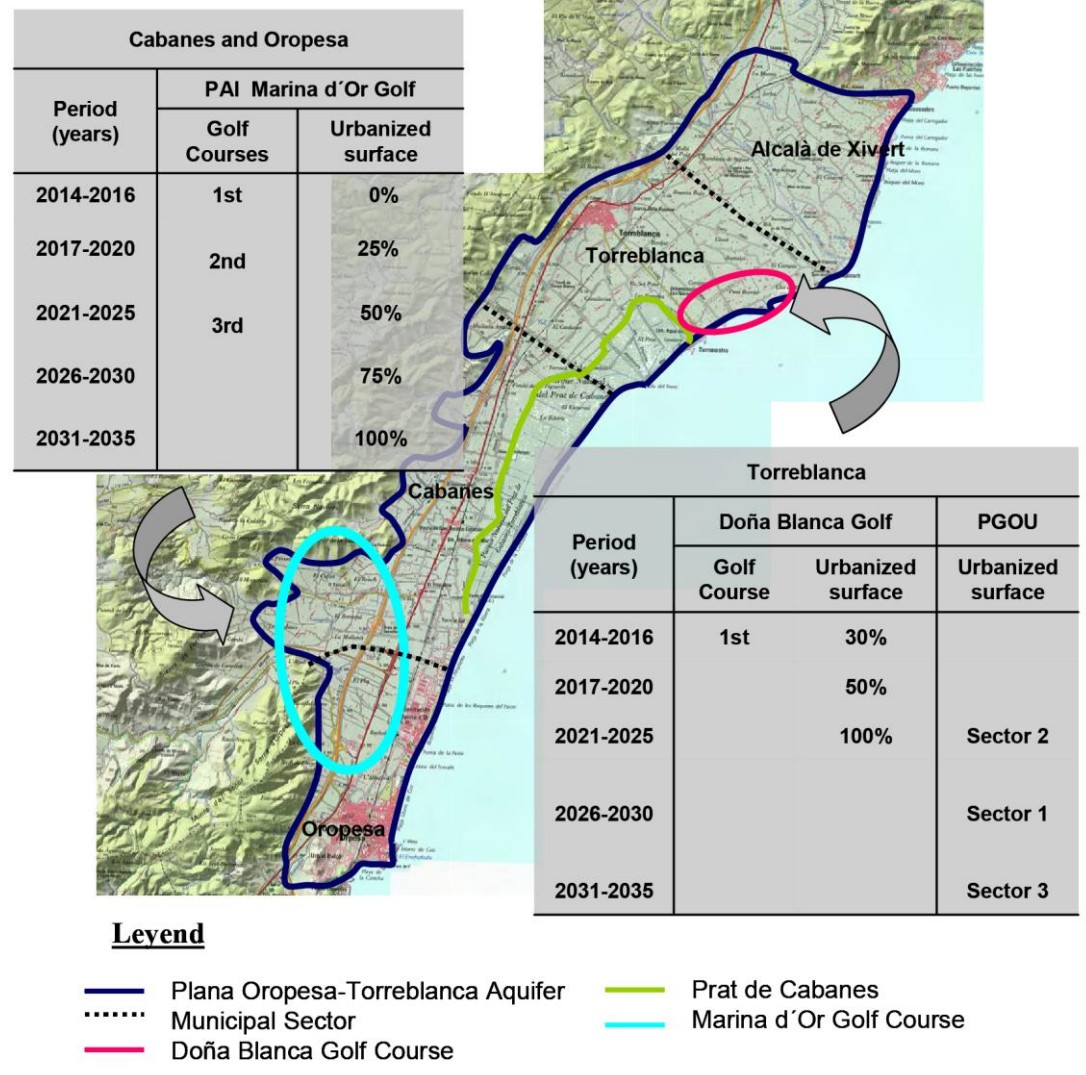

| Cabanes and Oropesa | | |
|---|---|---|
| Period (years) | PAI Marina d´Or Golf | |
| | Golf Courses | Urbanized surface |
| 2014-2016 | 1st | 0% |
| 2017-2020 | 2nd | 25% |
| 2021-2025 | 3rd | 50% |
| 2026-2030 | | 75% |
| 2031-2035 | | 100% |

| Torreblanca | | | |
|---|---|---|---|
| Period (years) | Doña Blanca Golf | | PGOU |
| | Golf Course | Urbanized surface | Urbanized surface |
| 2014-2016 | 1st | 30% | |
| 2017-2020 | | 50% | |
| 2021-2025 | | 100% | Sector 2 |
| 2026-2030 | | | Sector 1 |
| 2031-2035 | | | Sector 3 |

**Leyend**

— Plana Oropesa-Torreblanca Aquifer
····· Municipal Sector
— Doña Blanca Golf Course
— Prat de Cabanes
— Marina d´Or Golf Course

**Figure 5: Future land use scenarios in 2035.**

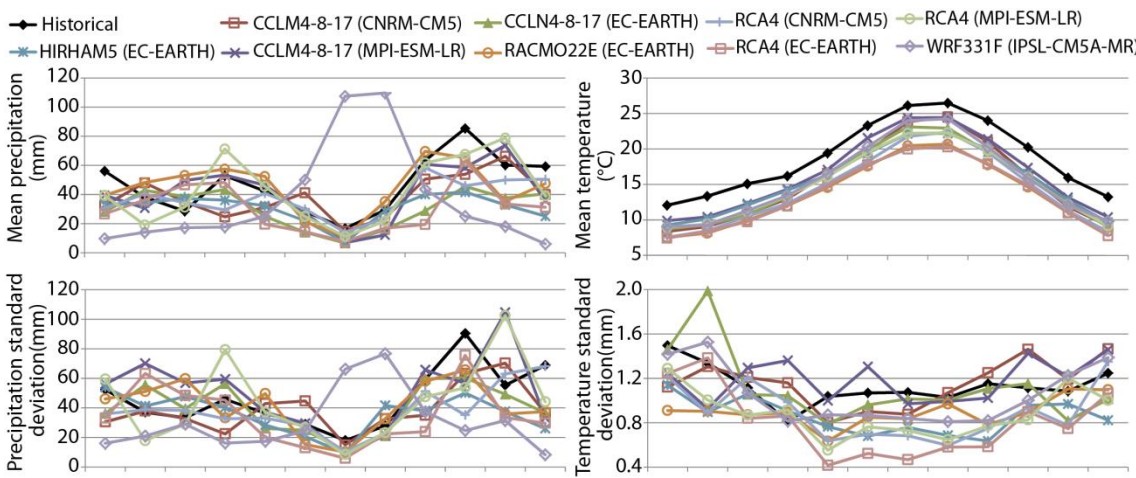

**Figure 6. Monthly mean and standard deviation of the historical and RCMs control series (rainfall and temperature) for the mean year in the period 1976-2000. RCMs data obtained from CORDEX project.**

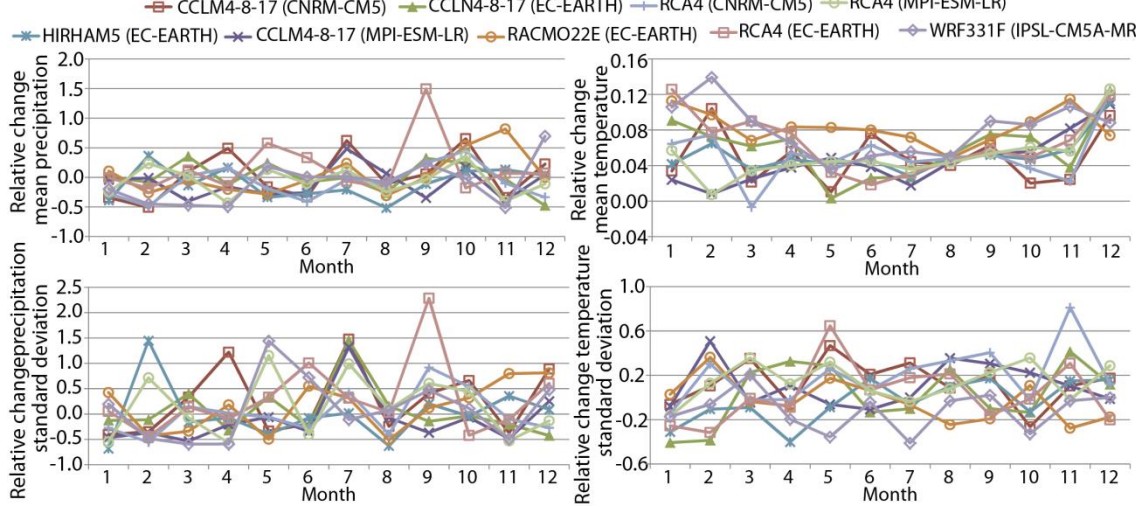

**Figure 7. Relative monthly change in mean and standard deviation of the future series (2011-2035) with respect to the control series (1976-2000) for the considered RCMs under the RCP8.5 emission scenario.**

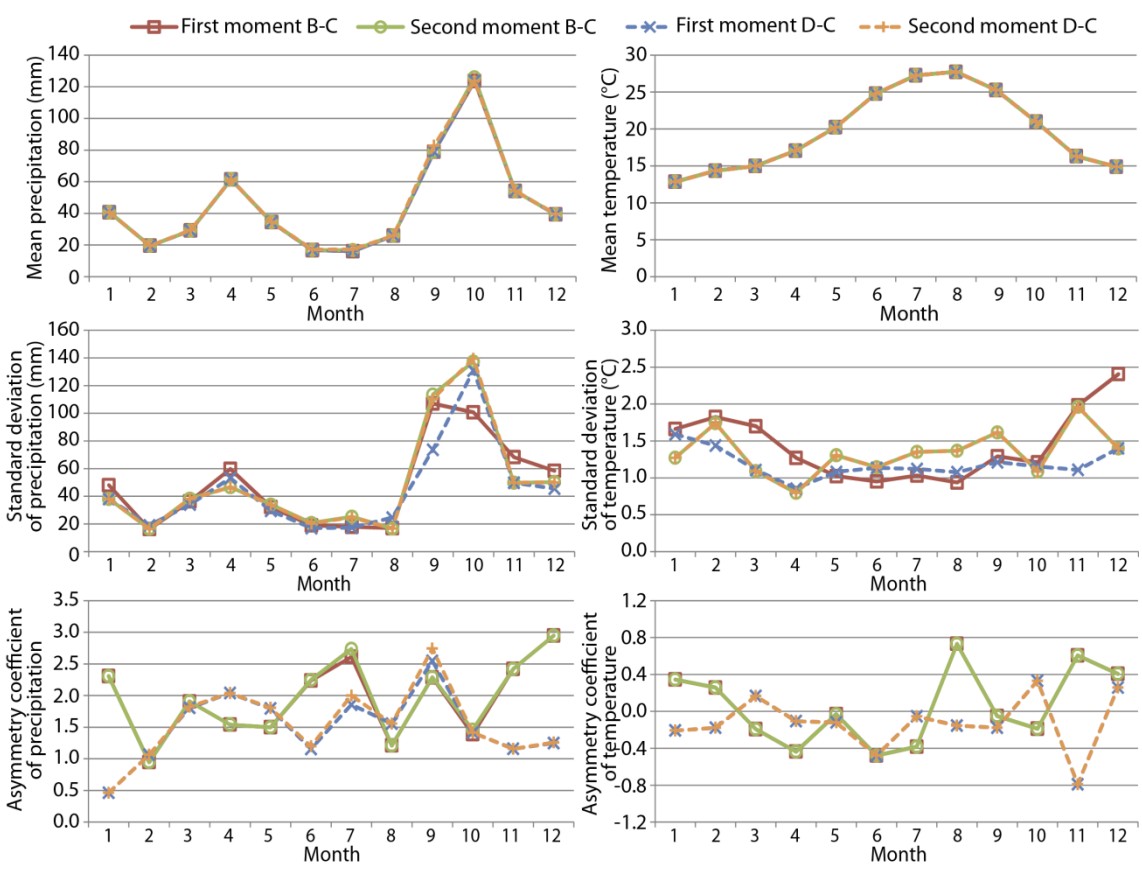

**Figure 8: Mean, standard deviation and asymmetry coefficients of future precipitation series (A) and future temperature series (B) for the average year for the RCM RCA4 model linked to the GCM CNRM-CM5.**

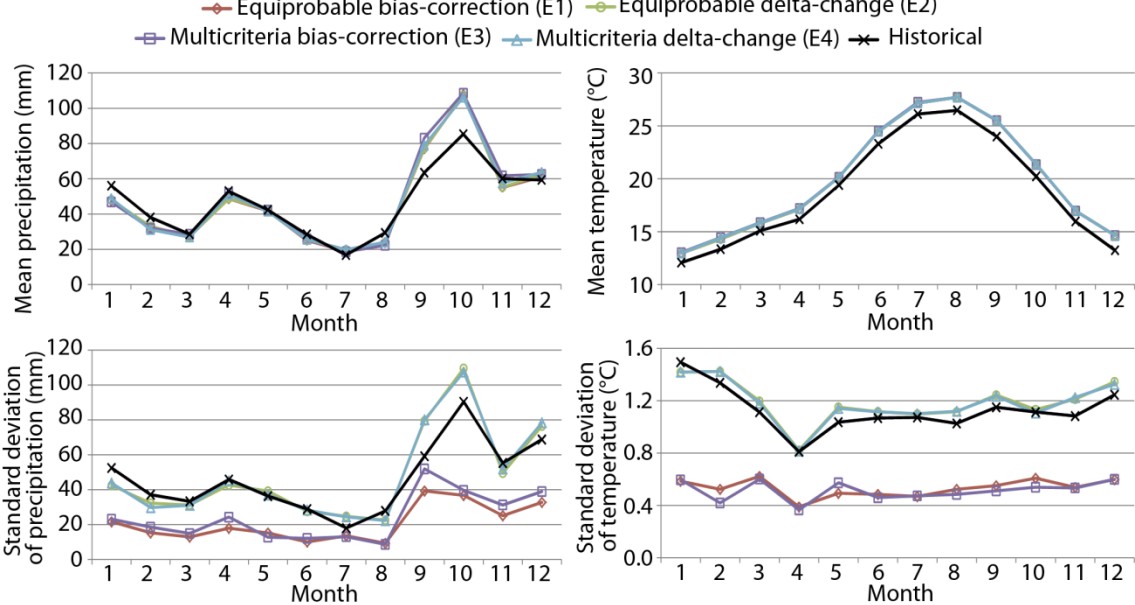

**Figure 9. Monthly mean and standard deviation of future precipitation and temperature series obtained by the four ensemble options.**

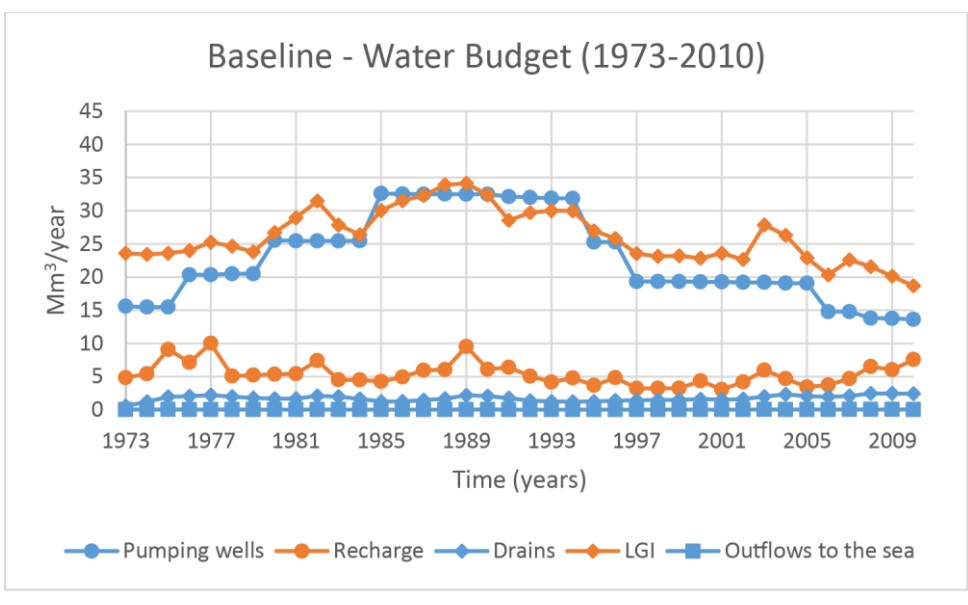

**Figure 10: Historical evolution of inflows and outflows in the aquifer.**

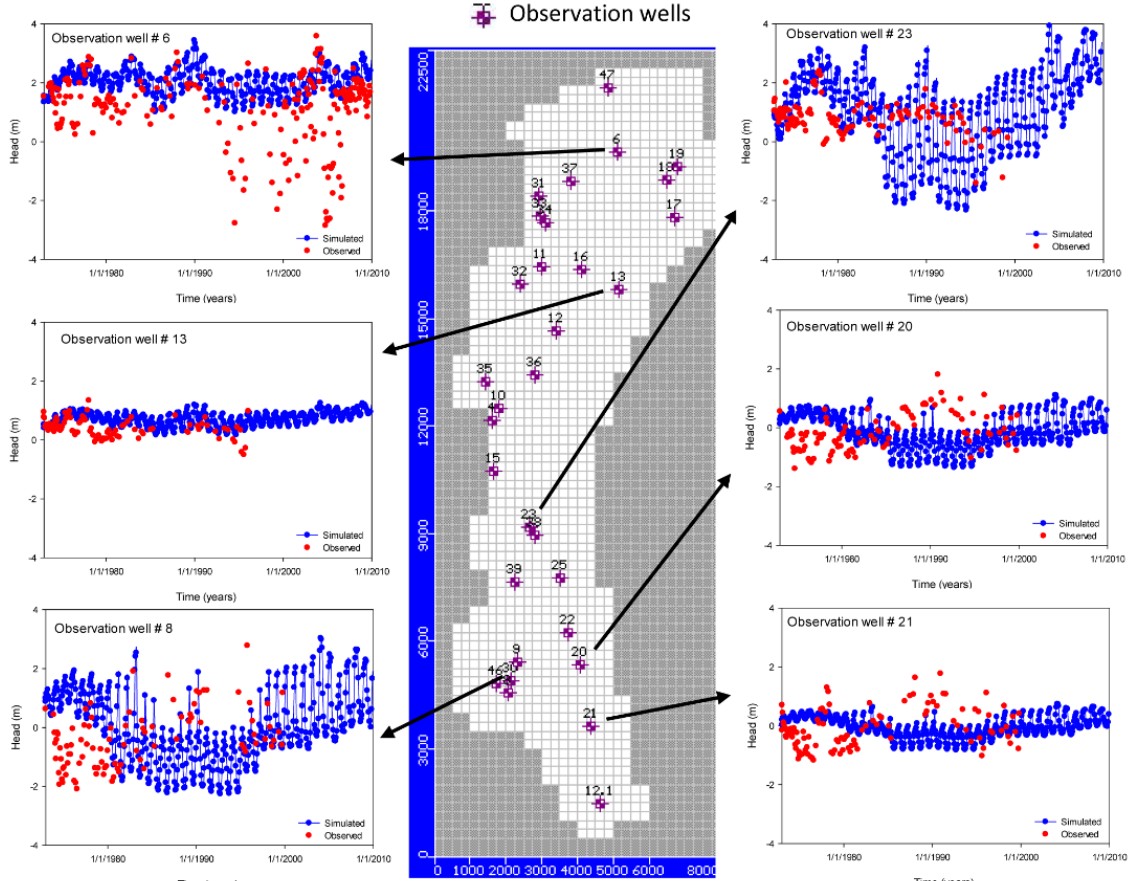

**Figure 11.a: Hydraulic head obtained with the models vs. data at some observation points.**

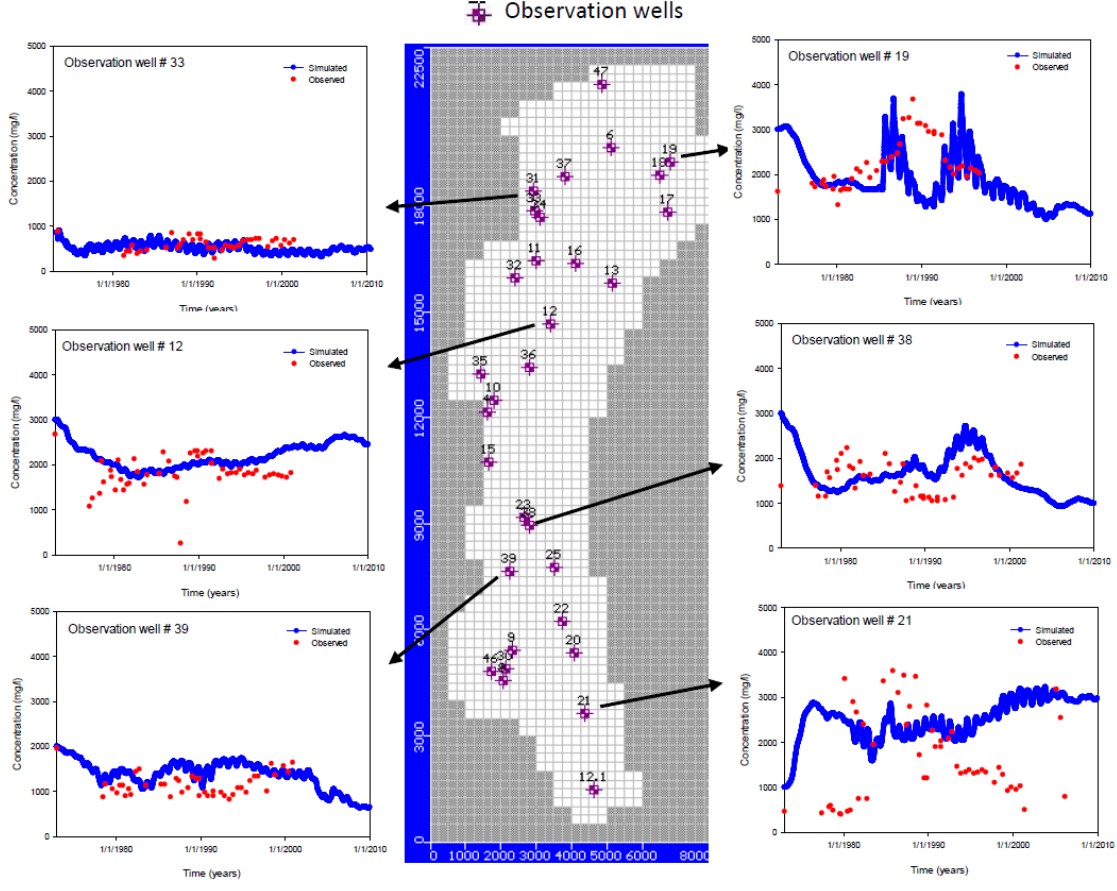

**Figure 11.b: Salinity obtained with the models vs. data at some observation points.**

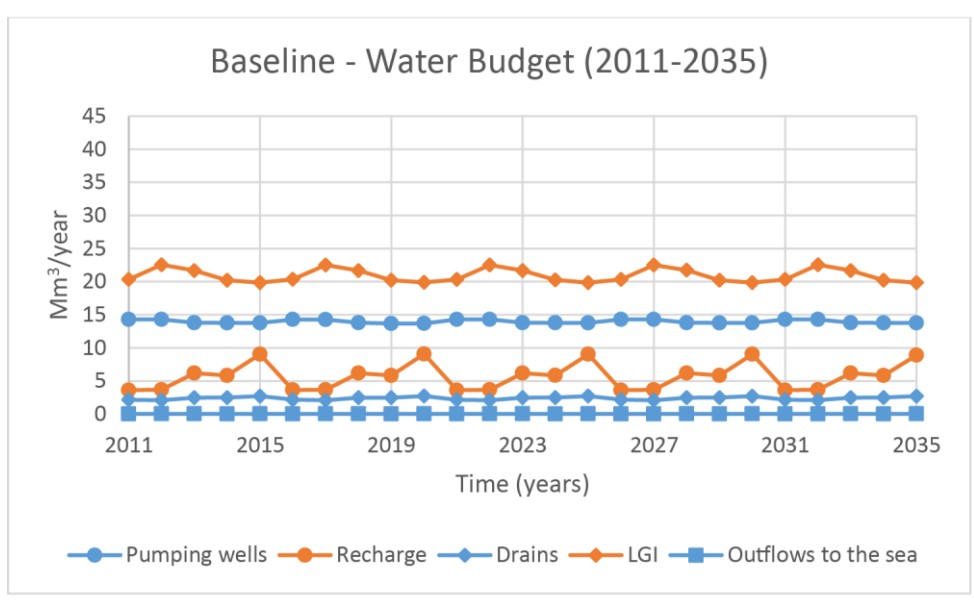

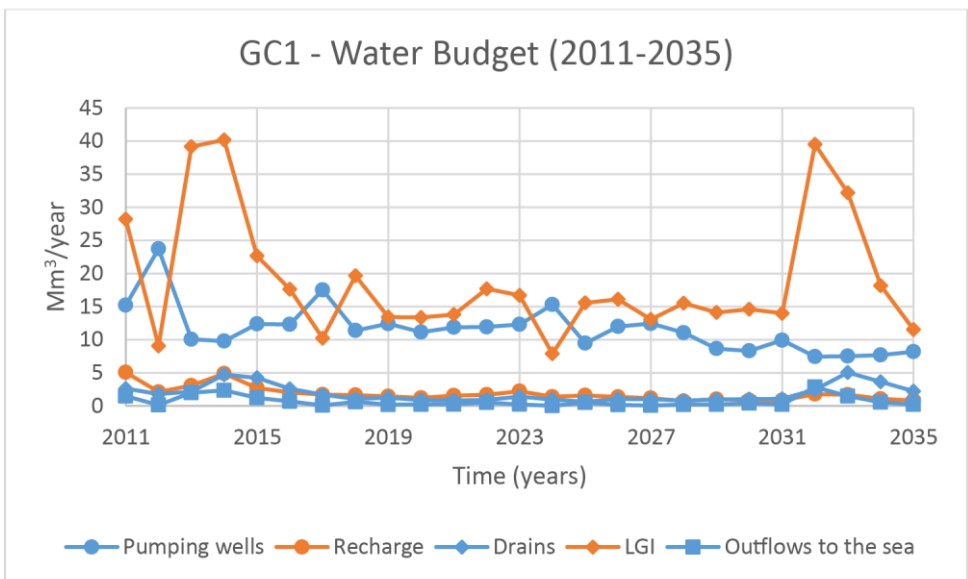

**Figure 12: Components of the flow budget evolution for the BL scenario and the GC1 scenario (horizon 2011-2035).**

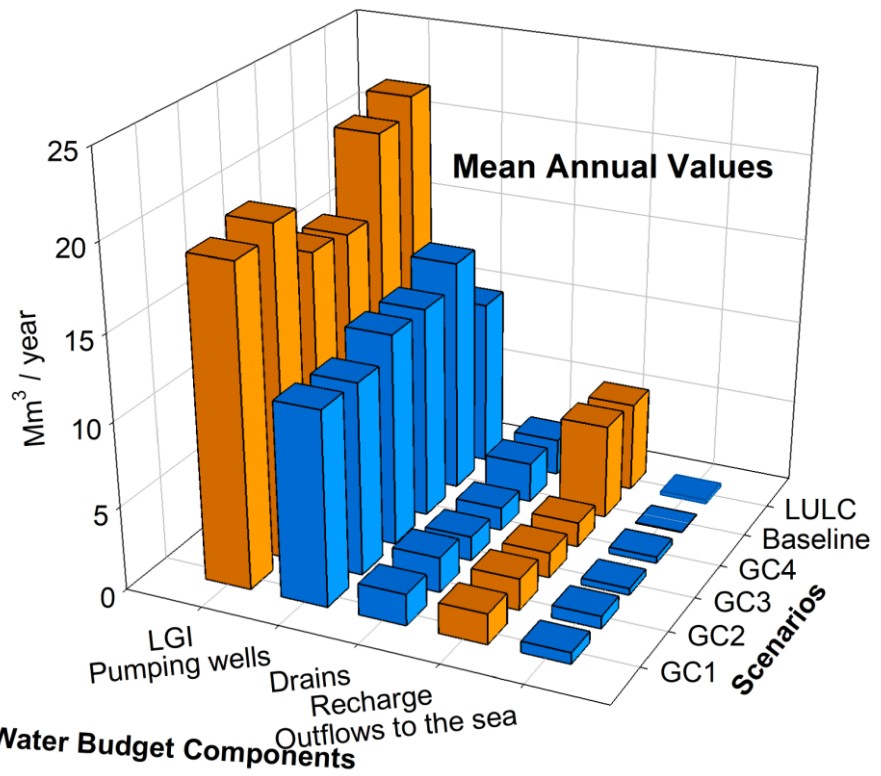

**Figure 13: Mean inflows and outflows for various global scenarios (GC1, GC2, GC3, GC4).**

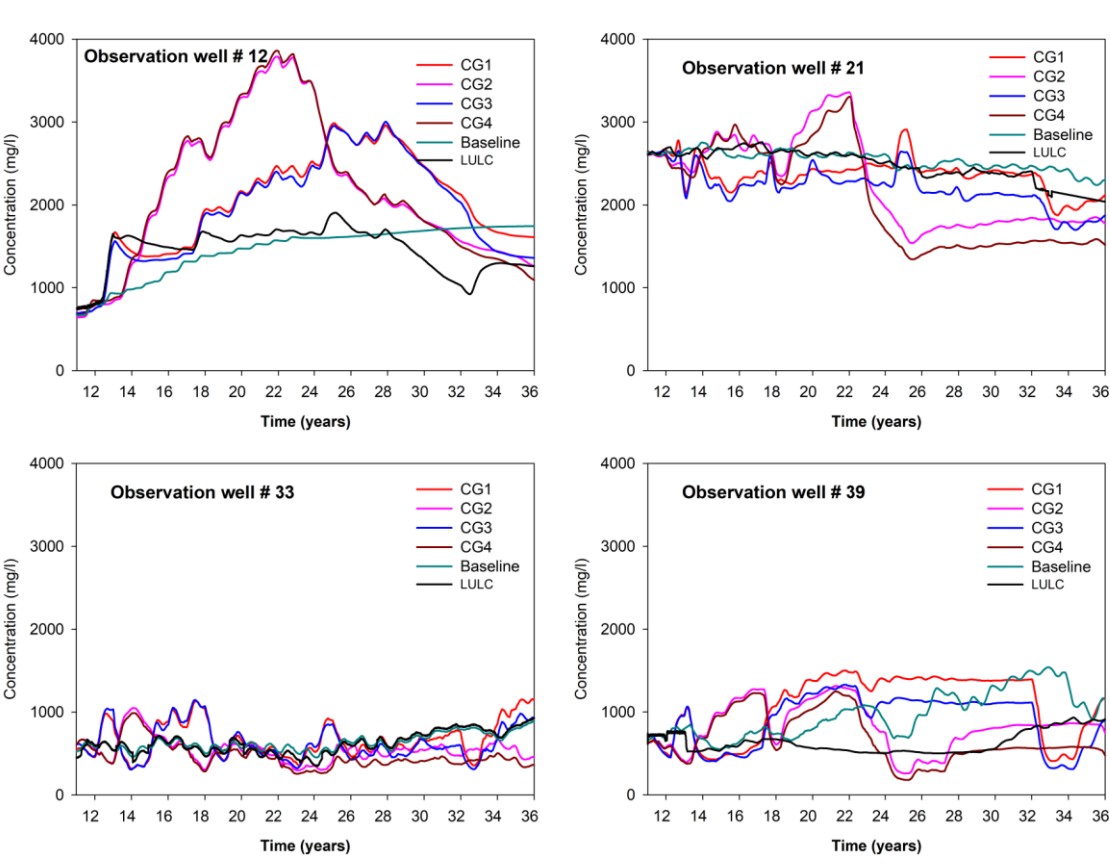

**Figure 14: Evolution of future salinity concentration at four observation points for the Baseline (BL) scenario and the four Global Change scenarios.**

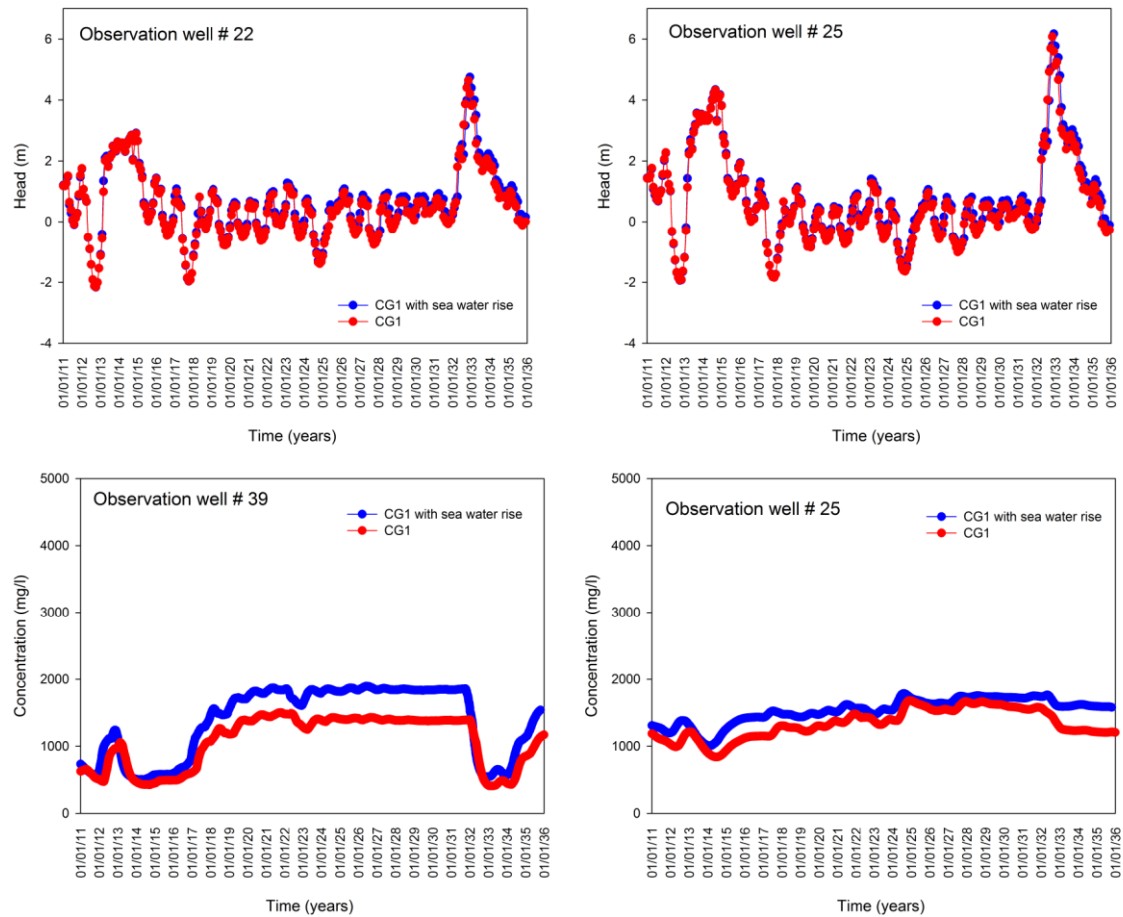

**Figure 15: Sensitivity of head and salinity obtained for GC1 to a SLR (0.19 m) scenario.**

| RCMs \ GCMs | CNRM-CM5 | EC-EARTH | MPI-ESM-LR | IPSL-CM5A-MR |
|---|---|---|---|---|
| CCLM4-8-17 | X | X | X | |
| RCA4 | X | X | X | |
| HIRHAM5 | | X | | |
| RACMO22E | | X | | |
| WRF331F | | | | X |

**Table 1: RCMs and GCMs considered.**

| ELIMINATED? | RCM | GCM |
|---|---|---|
| NO | CCLM4-8-17 | CNRM-CM5 |
| NO | CCLM4-8-17 | EC-EARTH |
| NO | CCLM4-8-17 | MPI-ESM-LR |
| NO | HIRHAM5 | EC-EARTH |
| NO | RACMO22E | EC-EARTH |
| NO | RCA4 | CNRM-CM5 |
| NO | RCA4 | EC-EARTH |
| YES | RCA4 | MPI-ESM-LR |
| NO | WRF331F | IPSL-CM5A-MR |

**Table 2: Eliminated and non-eliminated models in the multiple-criteria analysis**

| THRESHOLD 1% ELIMINATED? | RCM | GCM | TECHNIQUE |
|---|---|---|---|
| NO | CCLM4-8-17 | CNRM-CM5 | First moment |
| NO | CCLM4-8-17 | CNRM-CM5 | Second moment |
| YES | CCLM4-8-17 | EC-EARTH | First moment |
| NO | CCLM4-8-17 | EC-EARTH | Second moment |
| YES | CCLM4-8-17 | MPI-ESM-LR | First moment |
| YES | CCLM4-8-17 | MPI-ESM-LR | Second moment |
| NO | HIRHAM5 | EC-EARTH | First moment |
| NO | HIRHAM5 | EC-EARTH | Second moment |
| NO | RACMO22E | EC-EARTH | First moment |
| NO | RACMO22E | EC-EARTH | Second moment |
| YES | RCA4 | CNRM-CM5 | First moment |
| NO | RCA4 | CNRM-CM5 | Second moment |
| NO | RCA4 | EC-EARTH | First moment |
| NO | RCA4 | EC-EARTH | Second moment |
| YES | RCA4 | MPI-ESM-LR | First moment |
| NO | RCA4 | MPI-ESM-LR | Second moment |
| YES | WRF331F | IPSL-CM5A-MR | First moment |
| NO | WRF331F | IPSL-CM5A-MR | Second moment |

**Table 3: Eliminated and non-eliminated combinations of model and bias-correction technique.**