# Peer review of "Integrated assessment of future potential global change scenarios and their hydrological impacts in coastal aquifers. A new tool to analyse management alternatives in the Plana Oropesa-Torreblanca aquifer."

_Hydrology and Earth System Sciences, 2017_

## Referee Comment (RC1) · Anonymous Referee #1 · 12 Jun 2017

This paper defines a framework for assessing the effects of CC, LULC and SLR on a coastal aquifer where problems of saltwater intrusion are detected. Climate change models are integrated with LULC, SLR and double-density groundwater flow models in order to define future strategies for integrated water management in the study area.

The approach is ambitious and valid scientifically. Many models, however, are introduced but not clearly explained. Some models are described with excessive jargon

and others are barely defined and no reference is given (modified etr). As a result, the paper has a black box kind of content, which makes it very difficult to evaluate. I suggest that the authors upload some additional information that relates to the different models and steps they did in their work.

Another confusing point is the use of the acronym GC, GCi, GC1, GC2, etc. which are never correctly explained. The authors use also inconsistencies in defining concentration and relating it to density (density 1025 Kg/m3; salinity 1035 g/l), which can cause serious problems in the double-density models. It is not clear if in the models they use chloride concentration or salinity. Also the porosity used seems very small compared to the permeability detected in the aquifer. Another point that the authors do not address is the connection between the carbonate basement and the detritic aquifer.

Section 3.1.1 This section requires better explanation of the modeling (lines 20-30).

I find the correlation between observed and modeled hydraulic head and salinity very poor. Can the authors explain on which basis the results of the models are acceptable as a predictive tool?

The discussion and conclusion sections are very short and poorly quantitative and fail to point out how this kind of modeling can be used in integrated coastal water management. The authors should elaborate on their results and say explicitly how this knowledge can be used in an integrated water management framework of a coastal zone. Give also explicit examples of how this can be done.

Some specific points about the figures:

Figure 1: Vertical scale is missing in the figure. Not discussed in the text is the relationship between the carbonate rocks and the detritic aquifer. No explanation of the lithotype in the geologic time scale legend is given. There are too many eastings and northings in the map. Define them only at the corners of the figure. Confusing the color grey used for the aquifer and the Mediterrenean Sea.

Figure 2: The CORINE database is not mentioned in the text.

Figure 3: The overlap does not allow to distinguish well the data from the two watersheds. Also the choice of color is poor. Maybe use the same color for the same watershed.

Figure 4: It is not clear what sectors 1, 2 and 3 refer to in the text.

Figure 5: Please give also some information about the fact that you are presenting climate models data. This caption is not sufficient to understand what kind of data are presented.

Figure 6: See my note above. Also here some more information is needed. At least give the time frame for the climate change models.

Figure 7: I would have presented this figure much earlier on in the paper.

Figure 8: In wells 6, 23, 20, 8, and 21 there is a large difference between observed and modeled hydraulic head data. This, in a coastal context is not a good thing, because it makes the results of the double-density flow model unreliable. I think that the authors should address this large variability and explain how their flow model is still acceptable in view of this poor correlation.

Figure 8: I find the correlation between observed and modeled salinity very poor also here. Can the authors explain on which basis the results of the models are acceptable as a predictive tool.

Figure 9: It would be nice to separate the inflow from the outflow in this graph, so that it is clear the variation in the total yearly budget (you can do this by using the same color for inflows and different data point symbols; and a different color for outflows . with different data symbols).

Figure 11: Specify data are at monthly level.

Figure 12: See my note for Figure 9.

Figure 13: x axis should be "water budget components". Please specify a little bit better what the different CG's are. Hm3 / year is not a standard flow unit. Please specify.

Figure 14: A few words about well locations in the text would be helpful. Also, sometimes you talk about salinity and sometimes about chloride concentration. They are not the same thing. Could you please explain in the text what concentrations unit you are using and why?

I have attached a file with detailed requests for explanation in the text, some english corrections and suggestions. I hope this is helpful.

Please also note the supplement to this comment:
http://www.hydrol-earth-syst-sci-discuss.net/hess-2017-262/hess-2017-262-RC1-supplement.pdf

[Figure]

**Supplement:**

[revised manuscript text omitted]

---

## Referee Comment (RC2) · Anonymous Referee #2 · 5 Jul 2017

Unfortunately, the manuscript is not ready for publication yet. Below, a number of critical issues are raised, including methods, discussion and results.

Organization: - Chapter 2.3, 2.4 and 2.5 should be moved to Methods

Methods: - The applied modeling system is described as "integrated". However, there are no feed-backs in the system so it is misleading to call it integrated. A term like "coupled" would be more appropriate. - It is not clear how the rainfall-recharge model

was calibrated – which data and which period. Results on calibration missing. - It is not clear how spatial heterogeneity (it must be significant in this area) is handled. - Which area do the groundwater model cover? - Do the groundwater model describe both the Plioquaternary and the prequaternary formations? - Do the model take into account that the formations are fractured? - 11 model layers are used – is that sufficient to avoid too much numerical dispersion? - It is stated that inverse modeling is not used "due to the complexity of the case study dealt with". Does that imply that auto-calibration cannot be used for complex systems? If that is what you mean, please argue why. - As a minimum the match to the observations should be quantified by a few statistics (e.g., Mean Error, Root Mean Squared value) - Future climate signals are found by averaging the results from the available climate models and subsequently feed this averaged signal into the hydrological models. Alternatively, results from each individual climate model should have been used as input to the hydrological model system and averaged afterwards. Please document that the method used is appropriate. - Details on the downscaling methods completely missing. There are many versions of what you call "bias correction" – which one did you use? - How was the delta change method applied – monthly, yearly? Results: - The result section is very short and does actually not explain why the presented results are obtained. For example, why is the impact of sea level rise to insignificant? - What is most important – climate change or LULC changes?

Discussion: - There is no discussion of the results and this is critical. The manuscript cannot be published without a proper discussion of the results. This includes a comparison of methods and with results from other studies.

Uncertainty: - The uncertainty of the results are not touched at all. Considering the chain of model component that are used the total uncertainty of the obtained results must be significant. A discussion of this element is mandatory. Quantification would be even better.

262, 2017.

---

## Author Comment (AC1) · 2 Aug 2017

We appreciate the Editor for giving us the opportunity to improve the paper during the review process. Following the editor's suggestions we have replied each of the reviewers' comments. The answers are detailed in the attached file (hess-2017-262-supplement.pdf) as supplementary material. Based on this answer if the editor considers it appropriates we will prepare the revised version of the manuscript.

[Figure]

Please also note the supplement to this comment:
https://www.hydrol-earth-syst-sci-discuss.net/hess-2017-262/hess-2017-262-AC1-supplement.pdf

---

## Author Comment (AC2) · 2 Aug 2017

We thank the reviewer for the recognition of the interest of this research and for the comments formulated, which have helped us to realize that some aspects were not clearly stated in the original manuscript and could be improved in a new version of the manuscript. Following the reviewer suggestion we will upload some additional information that relates to the different models and steps we did in our work. The response

to his comments can be seen in the attached file (hess-2017-262-supplement.pdf) as supplementary material.

Please also note the supplement to this comment: https://www.hydrol-earth-syst-sci-discuss.net/hess-2017-262/hess-2017-262-AC2-supplement.pdf

---

## Author Comment (AC3) · 2 Aug 2017

Journal: Hydrol. Earth Syst. Sci.
Manuscript Number: hess-2017-262
Title: "Integrated assessment of future potential global change scenarios and their hydrological impacts in coastal aquifers. A new tool to analyse management alternatives under uncertainty in the Plana Oropesa-Torreblanca aquifer"
Special Issue: Assessing impacts and adaptation to global change in water resource systems depending on natural storage from groundwater and/or snowpacks

**\*\*\*\*\*\*\*\*\*\*\*\*\*\*\*\*\*\*\*\*\*\*\*\*\*\*\*\*\*\*\*\*\*\*\*\*\*\*\*\*\*\*\*\*\*\*\*\*\*\*\*\*\*\*\*\*\*\*\*\*\*\*\*\*\*\*\*\***

**Comments from the editors:**

**\*\*\*\*\*\*\*\*\*\*\*\*\*\*\*\*\*\*\*\*\*\*\*\*\*\*\*\*\*\*\*\*\*\*\*\*\*\*\*\*\*\*\*\*\*\*\*\*\*\*\*\*\*\*\*\*\*\*\*\*\*\*\*\*\*\*\*\***

**No more referee comments and short comments will be accepted. Now the public discussion shall be completed as follows:**

**You - as the contact author - are expected to publish final author comments on behalf of all co-authors no later than 02 Aug 2017 (final response phase). You are kindly requested to answer all referee comments and relevant short comments at: https://www.hydrol-earth-syst-sci-discuss.net/hess-2017-262/#discussion**

**To keep manuscript turnover times low we encourage you to submit your responses as soon as possible. Please note that your revised manuscript should not be prepared at this stage. Based on the responses, the Editor will be asked to take a decision about the further handling of your manuscripts.**

**Response to Comments from the Editors:**

*We appreciate the Editor for giving us the opportunity to improve the paper during the review process. Following the editor's suggestions we have replied each of the reviewers' comments. Based on this answer if the editor considers it appropriates we will prepare the revised version of the manuscript.*
* * *
**Reviewer #1 Comments to Author:**
* * *
1) **This paper defines a framework for assessing the effects of CC, LULC and SLR on a coastal aquifer where problems of saltwater intrusion are detected. Climate change models are integrated with LULC, SLR and double-density groundwater flow models in order to define future strategies for integrated water management in the study area. The approach is ambitious and valid scientifically. Many models, however, are introduced but not clearly explained. Some models are described with excessive jargon and others are barely defined and no reference is given (modified etr). As a result, the paper has a black box kind of content, which makes it very difficult to evaluate. I suggest that the authors upload some additional information that relates to the different models and steps they did in their work.**

*We thank the reviewer for the recognition of the interest of this research and for the comments formulated, which have helped us to realize that some aspects were not clearly stated in the original manuscript and could be improved in a new version of the manuscript. Following the reviewer suggestion we will upload some additional information that relates to the different models and steps we did in our work. More detail about it can be seen in the response to the specific comments.*

*For example, see the response to comment number 6, in which a more detailed description of the rainfall recharge model is proposed in order to understand and evaluate it.*

2) **Another confusing point is the use of the acronym GC, GCi, GC1, GC2, etc. which are never correctly explained.**

*Following the reviewer suggestion we will define each of the employed acronyms within the new version of the manuscript.*

*For example, in order to define GC, we would add the next sentence:*

We intent to analyze the impacts of Global Change (GC), which integrates impacts of Climate Change (CC) and Land Use and Land Cover (LULC) change.

*A detailed reviewed of the definition of each acronym will be performed in the new version of the manuscript.*

3) **The authors use also inconsistencies in defining concentration and relating it to density (density 1025 Kg/m3; salinity 1035 g/l), which can cause serious problems in the double-density models. It is not clear if in the models they use chloride concentration or salinity.**

*The salinity concentration of the seawater that we have used is 35 g/l in the SEAWAT simulations. We have used this value throughout all the revised version of the manuscript.*

**4) Also the porosity used seems very small compared to the permeability detected in the aquifer.**

*Sorry, there was a typo error in this paragraph. The specific yield takes values from 0.01 to 0.15 and effective porosity values from 0.01 to 0.13;*

*Nevertheless, in in order to explain more clearly the conceptual approach followed to define the model parameters we propose to add some sentences explaining it within the new version of the manuscript.*

We have replaced a highly heterogeneous porous media with an upscaled "equivalent" homogeneous porous media to represent the hydrogeological parameters since the cell size of the discretization is 250x250 m (e.g., Llopis-Albert and Capilla, 2010).
Then, we have used a value of the effective porosity based on available data, which were subsequently calibrated and upscaled using expert judgment. The results of the calibration process prove the worth of this approach.

-Llopis-Albert, C., Capilla, J.E. (2010). Stochastic inverse modelling of hydraulic conductivity fields taking into account independent stochastic structures: A 3D case study. Journal of Hydrology, 391(3–4), 277-288. DOI: 10.1016/j.jhydrol.2010.07.028.

**5) Another point that the authors do not address is the connection between the carbonate basement and the detritic aquifer.**

*Thank to the reviewer comment we have realized that we did not explain it properly within the manuscript.*
*The carbonate basement receives the contributions coming from the bordering aquifer (Maestrazgo aquifer) and feeds the detrital aquifer. The direction of the groundwater flow is from the carbonate basement to the detrital aquifer. The flow rate between both aquifers changes during the historical period with a minimum of 18 and a maximum of 34 Mm3/year as shows Figure 9 (see the Lateral Groundwater Inflows from the bordering aquifers (LGI)). It can be also appreciated in the next figure:*

[Figure]

**6) Section 3.1.1 This section requires better explanation of the modeling (lines 20-30).**

*In the next paragraphs we include a more detailed description of the rainfall-recharge model:*

Based on these historical climate and recharge data described in sections 2.2, we propose a simple empirical rainfall-recharge approach to generate yearly aquifer recharge series. Instead of defining an infiltration coefficient to deduce recharge directly from P data, which is a simple approximation commonly applied (Kirn et al., 2017), we propose a correction function by perturbing the historical series defined as the difference in P and E (hereafter referred to as PE series), modifying its mean and standard deviation to make them identical to the statistic of the historical aquifer recharge previously obtained from the lysimeter measurements, as follows:

$$R_i = PEn_i . \sigma_R + \overline{R} \tag{1}$$

where $R_i$ is the recharge series generated for the year i, $\sigma_R$ and $\overline{R}$ are the standard deviation and mean of the historical recharge series estimated using the infiltration rate coefficient obtained from previous lysimiter readings (Tuñon, 2000), and $Pn_i$ is the normalised historical PE series (P-E) obtained from:

$$Pn_i = \frac{PE_i - \overline{PE}}{\sigma_{PE}} \tag{2}$$

where $PE_i$ is the historical PE series for the year i, and $\sigma_{PE}$ and $\overline{PE}$ are the standard deviation and mean historical values of the series.

Taking the positive relationship of temperature (T) and actual evapotranspiration (E) into account (Arora, 2002; Gerrits et al., 2009), changes in T will determine the available water fraction for other balance components, including aquifer recharge. Different non-global

empirical models could be applied to assess the historical E from T series (e.g., Turc, 1954, 1961; Coutagne, 1954; Budyko, 1974; amongst others) as described in Arora (2002), Gerrits et al. (2009), and España et al. (2013). In this study, we applied Turc's model (1954, 1961), in which the results depend on mean annual T and solar irradiation over the latitude.

NEW REFERENCES:

Arora, V.K.: The use of the aridity index to assess climate change effect on annual runoff, Journal of Hydrology, 265, 164-177, 2002.

Budyko, M.I.: Climate and Life, Academic Press, New York, 508pp, 1974.

Coutagne, A.: 1954. Quelques considérations sur le pouvoir évaporant de l'atmosphere, le déficit d'écoulement effectif et le déficit d'écoulement maximum, La Houille Blanche, 360-369, 1954.

España, S., Alcalá, F.J., Vallejos, A., and Pulido-Bosch, A.: A GIS tool for modelling annual diffuse infiltration on a plot scale, Computers & Geosciences, 54, 318-325, 2013.

Gerrits, A.M.J., Savenije, H.H.G., Veling, E.J.M., Pfister, L.: Analytical derivation of the Budyko curve based on rainfall characteristics and a simple evaporation model. Water Resour. Res., 45, W04403, doi:10.1029/2008WR007308, 2009.

Kirn, L., Mudarra, M., Marín, A., Andreo, B., and Hartmann, A.: Improved Assessment of Groundwater Recharge in a Mediterranean Karst Region: Andalusia, Spain. In: Renard P., Bertrand C. (Eds.) EuroKarst 2016, Neuchâtel, Advances in Karst Science, 117-125, Springer, Cham, DOI: 10.1007/978-3-319-45465-8_13, 2017

Turc, L.: Water balance of soils: relationship between precipitation, evapotranspiration and runoff (in French), Ann Agron 5, 49-595 and 6, 5-131, 1954.

Turc, L.: Estimation of irrigation water requirements, potential evapotranspiration: A simple climatic formula evolved up to date, Annales Agronomiques, 12(1), 13-49, 1961.

*We also propose to modify section 2.2. in order to clarify the data that has been employed in the definition of the model.*

The calibration period is the same for the rainfall-recharge model than for the variable-density flow and transport model, which spans from 1973 to 2010. The data employed are historical climate data (rainfall and temperature) and the infiltration rate coefficient obtained from previous lysimiter readings from a neighbouring aquifer (Plana de Castellón; Tuñon, 2000). The mean historical recharge (85 mm/year) and its standard deviation (31 mm/year) obtained for this infiltration rate coefficient in the cited historical period is quite similar to the mean (89 mm/year) and standard deviation (27 mm/year) estimated by other authors who applied an atmospheric chloride mass balance (Alcalá and Custodio, 2014).

**7) I find the correlation between observed and modeled hydraulic head and salinity very poor. Can the authors explain on which basis the results of the models are acceptable as a predictive tool?**

*We agree with the reviewer, it would be desirable to have better adjustments between observed and modeled hydraulic head and salinity to have a greater confidence in the predictions made with the model. However, it is difficult to improve the approach proposed in this work for Plana de Oropesa-Torreblanca due to the following facts:*

*- The hydrogeological complexity of the aquifer makes it difficult to define a better approach taking into account its spatial heterogeneity, with fractured formations and preferential flow channels existing in the aquifer (Morell and Giménez, 1997). The spatial heterogeneity is handled by means of sequential indicator simulation using the computer code ISIM3D (Gómez-Hernández and Srivastava, 1990). See more details about how this heterogeneity is handled in the answer to the comments 5 and 7 of reviewer 2.*

*- The quality of some observation data is not very good. This problem is accentuated when considering the long simulated time period with historical data, which spans from 1973 to 2010. In this way, for a certain observation borehole there are data close in time with measurements quite disparate, which cannot be explained by any physical phenomenon. A statistical processing of data with expert judgment would be advisable to dismiss the wrong data. We have opted for using all available data (as a transparency measurement and in order to ease the reproduction of this exercise by other researchers).*

*- The lack of reliable estimates of dispersion coefficients (Naji et al., 1999) may prevent a proper adjustment for the Plana Oropesa-Torreblanca aquifer.*

*- A better fit might be achieved using a more refined spatial discretization, which would allow modeling the preferential flow channels existing in the aquifer (Morell and Giménez, 1997). However, the high computational burden when solving the variable-density flow and transport equations with SEAWAT prevents the use of a fine grid (e.g. Sreekanth and Datta, 2010).*

*- As a further research, we are intended to couple the salt water intrusion model with a management simulation–optimization model for control and remediation that would further prevent the use of a fine grid.*

*We have opted for using a variable-density model instead of sharp-interface solutions (most of them based on the Ghyben–Herzberg relation), which have been extensively employed to define management models because of its simplicity in terms of required parameters and computational burden (e.g., Mantoglou et al., 2004). This is because they better describe the dynamic of real complex coastal aquifers despite the limitations in the available data and the course discretization used for the Plana Oropesa-Torreblanca aquifer. These drawbacks lead to some differences between observed and simulated data, especially in the salinity concentration.*

*Nevertheless, although due to the differences between observed and simulated data the uncertainty of the prediction coming from the model grows and we have to be be cautious with the conclusions obtained in the application to the case study (aspect included in the limitation section), we think that the fit is good enough to capture the general trend of the hydraulic head and salinity variables within a quite long calibration period (37 years, from 1973 to 2010), and therefore, to assess general impacts of climate and land-use and land-cover changes. Instead of using a sharp-interface solution that does not take into account the diffusion and dispersion mechanisms we propose a more physical approach to approximate the dynamic of the salinity concentration.*

*In order to clarify these issues within the new version of the manuscript we intend to add a **limitation section,** in which all these issues and considerations would be added and highlighted. On the other hand, from a methodological point of view, as the reviewer pointed in his first comments, the proposed approach is ambitious and valid scientifically, which is the main scope of this research work.*

**8) REFERENCES:**

*\*Mantoglou A, Papantoniou M, Giannoulopoulos P. 2004. Management of coastal aquifers based on nonlinear optimization and evolutionary algorithms. Journal of Hydrology 297(1-4): 209 228. DOI: 10.1016/j.jhydrol.2004.04.011*

*\*Morell, I. and Giménez, E.: Hydrogeochemical analysis of salinization processes in the coastal aquifer of Oropesa (Castellón, Spain), Environmental Geology, 29(1/2), 118-131, 1997.*

*\*Naji A, Cheng AD, Quazar D. 1999. BEM solution of stochastic seawater intrusion problems. Engineering Analysis with Boundary Elements 23: 529–37.*

*\*Sreekanth J, Datta B. 2010. Multi-objective management of saltwater intrusion in coastal aquifers using genetic programming and modular neural network based surrogate models. Journal of Hydrology 393: 245–256.*

9) **The discussion and conclusion sections are very short and poorly quantitative and fail to point out how this kind of modeling can be used in integrated coastal water management. The authors should elaborate on their results and say explicitly how this knowledge can be used in an integrated water management framework of a coastal zone. Give also explicit examples of how this can be done.**

*Following his suggestion we will do our best to improve the discussion and the conclusion section within the revised version of the manuscript.*

*The proposed approach allows to assess the impacts of different climatic and land uses change scenarios in terms of global flow balance, as well as a distributed approximation of the hydraulic head and salinity. The components of the global balance for the simulated future scenarios show that, in general terms, the intrusion problems will not grow and will even be*

*reduced slightly, as the outflows to the sea will not decrease, due in part to the reduction and redistribution of pumping in the mentioned scenarios (see Figure 13). From the results in terms of salinity at specific observation points, we can identify areas where the situation will clearly improve throughout the future horizon contemplated with the proposed scenarios. For example, at observation point 21 (in the southern area) the salinity would be reduced with the contemplated scenarios. In other areas, around the observation point 12, the situation deteriorates significantly until arriving at the last years of the horizon 2035.*

*As commented above the expected results without considering global change impacts are likely to be too pessimistic or optimistic, depending on the location. These results can be useful to the authorities in charge of the planning and definition of management policies in the Plana de Oropesa Torrablanca. Modifying the inpust of the integrated modeling framework developed it could be useful to assess potential effect of adaptation measures to global change. Participatory processes including the relevant stakeholders might be essential in the successful definition of adaptation measures for groundwater management (Pulido-Velazquez et al., 2015) and this modeling framework could be useful in the search for consensus - "shared vision" models.*

*On the other hand, in order to perform a quantitative analysis and discussion of the relative significance of climate change and LULC in the final impacts we will include the results of an additional scenario that we are currently simulating which considers future LULC assuming that there is not climate change.*

*We will also discuss the methods and results comparing with other previous approaches and studies. In this answer we include examples of some of the changes that would be introduced in the future manuscript.*

*For example, from a methodological point of view, the proposed approach has some similarities with that proposed by Pulido-Velazquez et al. (2015), in which an integrated analysis of global change is performed including climate and LULC changes. The most important difference is that in this case a coastal aquifer is studied and a variable density model is used to propagate the impacts on it.*

*For example, with respect to the sensitivity of the SLR on the results, we find in the literature other examples in which the sensitivity of seawater intrusion to the SLR would be low. Chan et al. (2011) obtained this conclusion in a synthetic confined coastal aquifer in which recharge in unchanged; Rasmussen et al. (2013) obtained the same conclusion for an inland coastal aquifer with minor SLRs. In our case the maximum value of SLR considered is 0.19m (in 2035), and it is quite low with respect to the level fluctuations experienced in most of the observation wells (see Figure 15). For this reason the sensitivity of the flow and transport is low. Nevertheless other*

*authors, as Werner and Simmons (2009) showed that in unconfined aquifers the influence of the inland boundary condition can be significant to its sensitivity to SLR.*

*Finally we will also summarize pros and cons of our study within a new subsection included in the discussion, a "limitation of the results and future research works" subsection. For example, in the **limitation** we would explain why it is difficult to improve the approach proposed for Plana de Oropesa-Torreblanca although it would be desirable to have better adjustments between observed and modeled hydraulic head and salinity to have a greater confidence in the model predictions (see response to question 7). Nevertheless, although due to the differences between observed and simulated data the uncertainty of the prediction coming from the model grows and we have to be cautious with the conclusions obtained in the application to the case study (aspect included in the limitation section), we think that the fit is good enough to capture the general trend of the hydraulic head and salinity variables within a quite long calibration period (37 years, from 1973 to 2010), and therefore, to assess general impacts of climate and land-use and land-cover changes. On the other hand, from a methodological point of view, as the reviewer 1 pointed in his first comments, the proposed approach is ambitious and valid scientifically, which is the main scope of this research work. As we show in the answer to the second reviewer comment, although the assessment of uncertainty out of the scope of the present paper, a proper analysis of it could be performed in **future research works**.*

**10) SOME SPECIFIC POINTS ABOUT THE FIGURES:**

**Figure 1: Vertical scale is missing in the figure. Not discussed in the text is the relationship between the carbonate rocks and the detritic aquifer. No explanation of the lithotype in the geologic time scale legend is given. There are too many eastings and northings in the map. Define them only at the corners of the figure. Confusing the color grey used for the aquifer and the Mediterrenean Sea.**

*Following the reviewer suggestion we have updated Figure 1.*

[Figure]

**Figure 2: The CORINE database is not mentioned in the text.**

*Following the reviewer suggestion we have mentioned it within the new version of the manuscript.*

**Figure 3: The overlap does not allow to distinguish well the data from the two watersheds. Also the choice of color is poor. Maybe use the same color for the same watershed.**

*We have eliminated the overlap in the monthly data and we have changed other aspects to clarify the figure.*

[Figure]

**Figure 5: Please give also some information about the fact that you are presenting climate models data. This caption is not sufficient to understand what kind of data are presented.**

*Done, we have modified the figure caption.*

Figure 5. Monthly mean and standard deviation of the historical and RCMs control series (rainfall and temperature) for the mean year in the period 1976-2000. RCMs obtained from CORDEX project.

**Figure 6: See my note above. Also here some more information is needed. At least give the time frame for the climate change models.**

*Done, we have modified the figure caption.*

Figure 6. Relative monthly change in mean and standard deviation of the future series (2011-2035) with respect to the control series (1976-2000) for the considered RCMs under the RCP8.5 emission scenario.

**Figure 7: I would have presented this figure much earlier on in the paper.**

*Following the reviewer suggestion we will include it earlier in the new version of the manuscript. This reviewer comment is also linked with the comment number 2 of the reviewer 2 about the organization of the manuscript (Chapter 2.3, 2.4 and 2.5 should be moved to Methods)*

**Figure 8: In wells 6, 23, 20, 8, and 21 there is a large difference between observed and modeled hydraulic head data. This, in a coastal context is not a good thing, because it makes the results of the double-density flow model unreliable. I think that the authors should address this large variability and explain how their flow model is still acceptable in view of this poor correlation.**

*See the answer to comment number 7*

**Figure 8: I find the correlation between observed and modeled salinity very poor also here. Can the authors explain on which basis the results of the models are acceptable as a predictive tool.**

*See the answer to comment number 7*

**Figure 9: It would be nice to separate the inflow from the outflow in this graph, so that it is clear the variation in the total yearly budget (you can do this by using the same color for inflows and different data point symbols; and a different color for outflows . with different data symbols).**

*Following the reviewers' comment we have separated the inflows (orange) and the outflows (blue) in Fig. 9 and 12.*

[Figure]

**Fig. 9**

**Figure 11: Specify data are at monthly level.**

*Done, we have provided the information in the figure caption.*

**Figure 11.** Monthly mean and standard deviation of future precipitation and temperature series obtained by the four ensemble options.

**Figure 12: See my note for Figure 9.**

*As in Figure 9, we have separated the inflows and the outflow in the graph.*

[Figure]

[Figure]

**Fig. 12**

**Figure 13: x axis should be "water budget components". Please specify a little bit better what the different CG's are. Hm3 / year is not a standard flow unit. Please specify.**

*Sorry for the mistake. It is a typo error. It should be GC scenarios (global change scenarios) instead of CG. We have corrected it. The units are Millions of cubic meters per year (Mm3/year), we will also correct it in the new version.*

[Figure]

**Figure 13**

**Figure 14: A few words about well locations in the text would be helpful. Also, sometimes you talk about salinity and sometimes about chloride concentration. They are not the same thing. Could you please explain in the text what concentrations unit you are using and why?**

*Following the reviewer suggestion we will include some words describing the location of the wells represented in Figure 14. The evolution in four wells roughly equispaced were represented to cover the extension of the aquifer from north to south. In order to follow it more easily we have ranked the graphics included in Figure 14, starting with the more northerly and moving towards the south (observation wells 33, 12, 39 and 31 respectively).*

*On the other hand, as the reviewer points out us we have used both terms indistinctly in the text and we have made a mistake when using it within the Figure 14 caption. Instead of chloride concentration it should be salinity concentration. We have used both terms within the manuscript because data is provided as chloride concentration, while in SEAWAT simulations we use salinity (mg/l) as concentration unit. The conversion is performed according to the following equation (e.g., Williams and Sherwood, 1994):*

$$S\ (^o/_{oo}) = 1.80655 \times Cl\ (^o/_{oo})$$

*where S is salinity and Cl- is Chlorinity.*

*Williams, W.D., Sherwood, J.E.(1994). Definition and measurement of salinity in salt lakes. International Journal of Salt Lake Research 3(1), 53–63.*

**11) I have attached a file with detailed requests for explanation in the text, some English corrections and suggestions. I hope this is helpful.**

*We sincerely appreciate the annotations provided by the reviewer in the attached file as complementary material. It highlights the paragraphs and sentences that are not clear enough and should be modified and improved within the new version of the manuscript. It will be really helpful to improve the clarity of the exposition in the new version of the manuscript. Each of the comments will be taken into account.*

**\*\*\*\*\*\*\*\*\*\*\*\*\*\*\*\*\*\*\*\*\*\*\*\*\*\*\*\*\*\*\*\*\*\*\*\*\*\*\*\*\*\*\*\*\*\*\*\*\*\*\*\*\*\*\*\*\*\*\*\*\*\*\*\*\*\*\*\*\*\***
**Reviewer #2 Comments to Author:**
**\*\*\*\*\*\*\*\*\*\*\*\*\*\*\*\*\*\*\*\*\*\*\*\*\*\*\*\*\*\*\*\*\*\*\*\*\*\*\*\*\*\*\*\*\*\*\*\*\*\*\*\*\*\*\*\*\*\*\*\*\*\*\*\*\*\*\*\*\*\***

**1) Unfortunately, the manuscript is not ready for publication yet. Below, a number of critical issues are raised, including methods, discussion and results.**

*We will try to do our best to improve the manuscript following the valuable comments from both reviewers.*

**2) Organization: Chapter 2.3, 2.4 and 2.5 should be moved to Methods**

*Following the reviewers' comment we will move those chapters to the Method section.*

**3) Methods: The applied modeling system is described as "integrated". However, there are no feed-backs in the system so it is misleading to call it integrated. A term like "coupled" would be more appropriate.**

*Following the reviewers' comment we will change the term "integrated" to "coupled" throughout all the manuscript.*

**4) It is not clear how the rainfall-recharge model was calibrated – which data and which period. Results on calibration missing.**

*Thank to the reviewer comment we have realized that the rainfall recharge model needed a more detailed and clear description within the manuscript. In order to explain it more properly we propose to modify Section 2.2 (about the historical data) and section 3.1.1 (rainfall Recharge model):*

*The calibration period is the same for the rainfall-recharge model than for the variable-density flow and transport model, which spans from 1973 to 2010. The data employed are historical climate data (rainfall and temperature) and the infiltration rate coefficient obtained from previous lysimiter readings from a neighbouring aquifer (Plana de Castellón; Tuñon, 2000). The mean historical recharge (85 mm/year) and its standard deviation (31 mm/year) obtained for this infiltration rate coefficient in the cited historical period is quite similar to the mean (89 mm/year) and standard deviation (27 mm/year) estimated by other authors who applied an atmospheric chloride mass balance (Alcalá and Custodio, 2014).*

Based on these historical climate and recharge data described in sections 2.2, we propose a simple empirical rainfall-recharge approach to generate yearly aquifer recharge series. Instead of defining an infiltration coefficient to deduce recharge directly from P data, which is a simple approximation commonly applied (Kirn et al., 2016), we propose a correction function by perturbing the historical series defined as the difference in P and E (hereafter referred to as PE series), modifying its mean and standard deviation to make them identical to the statistic of the historical aquifer recharge previously obtained from the lysimeter measurements, as follows:

$$R_i = PEn_i . \sigma_R + \overline{R} \tag{1}$$

where $R_i$ is the recharge series generated for the year i, $\sigma_R$ and $\overline{R}$ are the standard deviation and mean of the historical recharge series estimated using the infiltration rate coefficient obtained from previous lysimiter readings (Tuñon, 2000), and $Pn_i$ is the normalised historical PE series (P-E) obtained from:

$$Pn_i = \frac{PE_i - \overline{PE}}{\sigma_{PE}} \tag{2}$$

where $PE_i$ is the historical PE series for the year i, and $\sigma_{PE}$ and $\overline{PE}$ are the standard deviation and mean historical values of the series.

Taking the positive relationship of temperature (T) and actual evapotranspiration (E) into account (Arora, 2002; Gerrits et al., 2009), changes in T will determine the available water fraction for other balance components, including aquifer recharge. Different non-global empirical models could be applied to assess the historical E from T series (e.g., Turc, 1954, 1961; Coutagne, 1954; Budyko, 1974; amongst others) as described in Arora (2002), Gerrits et al. (2009), and España et al. (2013). In this study, we applied Turc's model (1954, 1961), in which the results depend on mean annual T and solar irradiation over the latitude.

*Next Figure shows the historical yearly evolution of the rainfall recharge in the aquifer obtained with the calibrated models:*

[Figure]

NEW REFERENCES:

Arora, V.K.: The use of the aridity index to assess climate change effect on annual runoff, Journal of Hydrology, 265, 164-177, 2002.

Budyko, M.I.: Climate and Life, Academic Press, New York, 508pp, 1974.

Coutagne, A.: 1954. Quelques considérations sur le pouvoir évaporant de l'atmosphere, le déficit d'écoulement effectif et le déficit d'écoulement maximum, La Houille Blanche, 360-369, 1954.

España, S., Alcalá, F.J., Vallejos, A., and Pulido-Bosch, A.: A GIS tool for modelling annual diffuse infiltration on a plot scale, Computers & Geosciences, 54, 318-325, 2013.

Gerrits, A.M.J., Savenije, H.H.G., Veling, E.J.M., Pfister, L.: Analytical derivation of the Budyko curve based on rainfall characteristics and a simple evaporation model. Water Resour. Res., 45, W04403, doi:10.1029/2008WR007308, 2009.

Kirn, L., Mudarra, M., Marín, A., Andreo, B., and Hartmann, A.: Improved Assessment of Groundwater Recharge in a Mediterranean Karst Region: Andalusia, Spain. In: Renard P., Bertrand C. (Eds.) EuroKarst 2016, Neuchâtel, Advances in Karst Science, 117-125, Springer, Cham, DOI: 10.1007/978-3-319-45465-8_13, 2017

Turc, L.: Water balance of soils: relationship between precipitation, evapotranspiration and runoff (in French), Ann Agron 5, 49-595 and 6, 5-131, 1954.

Turc, L.: Estimation of irrigation water requirements, potential evapotranspiration: A simple climatic formula evolved up to date, Annales Agronomiques, 12(1), 13-49, 1961.

**5) It is not clear how spatial heterogeneity (it must be significant in this area) is handled. –**

*Following the reviewer's suggestion we will describe it with more detail within the revised version of the manuscript.*

*The spatial heterogeneity is tackled using the concept of multiple statistical populations (Llopis-Albert and Capilla, 2010), in which the rock matrix and each fracture is represented as independent statistical population. The random function for each structure (i.e., the aquifer matrix and fractures) is modeled based on a geostatistical analysisconditionated to its own statistical distribution (i.e., hydraulic conductivity data as well as geological information and expert judgment). The random function is supposed to be as MultiGaussian for the rock matrix, while the fractures are considered as non-MultiGaussian. In this way, the rock matrix is generated by sequential Gaussian simulation using the code GCOSIM3D (Gómez-Hernández and Srivastava, 1990), while the fractures are generated by sequential indicator simulation using the code ISIM3D (Gómez-Hernández and Journel, 1993). The latter code makes use of local conditional cumulative density functions (ccdfs) defined by conductivity measurements and the corresponding indicator variograms. Therefore, the spatial heterogeneity is modelled as an equivalent porous media (e.g., Llopis-Albert and Capilla, 2010). On the one hand, the hydraulic conductivity data for fractures presents high values, greater than 1000 m/d. This allows the reproduction of strings of extreme values of hydraulic conductivity that often take place in nature and can be crucial in order to obtain realistic and safe estimations of mass transport predictions. That is, it allows reproducing preferential flow channels in strongly heterogeneous aquifers or fractured formations. On the other hand, the hydraulic conductivity data for the aquifer matrix cover a wide range of values, i.e., from 5 to 200 m/d. In addition, for each cell we have defined a vertical hydraulic conductivity equals to a tenth of the horizontal hydraulic conductivity.*

*The position of fractures is deterministically incorporated in the model based on geological information and expert judgment, thus allowing to classify the cell models. Those cells of the model intersected by a fracture are assigned conductivities according to the intersecting fracture, and those that are not are considered as cells belonging to the rock matrix.*

*REFERENCES:*

*Llopis-Albert, C., Capilla, J.E. (2010). Stochastic inverse modelling of hydraulic conductivity fields taking into account independent stochastic structures: A 3D case study. Journal of Hydrology, 391(3–4), 277-288. DOI: 10.1016/j.jhydrol.2010.07.028.*
*-Gómez-Hernández, J. J., and Journel, A. G. (1993). "Joint simulation of MultiGaussian random variables." Geostatistics tróia´92, A. Soares, ed., Vol. 1, Kluwer, Dordrecht, The Netherlands, 85–94.*
*-Gómez-Hernández, J.J., Srivastava, R.M. (1990). ISIM3D: An ANSI-C three dimensional multiple indicator conditional simulation program. Computer and Geosciences 16 (4), 395–440.*

**6) Which area do the groundwater model cover? Do the groundwater model describe both the Plioquaternary and the prequaternary formations?**

*The groundwater flow model describes both formations. The K data cover both formations, so that the K field obtained using the ISIM3D code also takes them into account.*

**7) Do the model take into account that the formations are fractured?**

*Note that one of the main advantages of the ISIM3D code is that it does not require assuming the classical multiGaussian hypothesis, which allows the reproduction of strings of extreme values of K that often take place in nature and can be crucial in order to obtain realistic and safe estimations of mass transport predictions. That is, it allows reproducing preferential flow channels in strongly heterogeneous aquifers or fractured formations. Therefore the model takes into account the fractured formations. Following the reviewers' comment this considerations have been added to the manuscript.*

**8) 11 model layers are used – is that sufficient to avoid too much numerical dispersion?**

*Note that we need to balance the use of a more refined discretization (in order to reduce the numerical dispersion) and the computational burden when solving the variable-density flow and transport equations with SEAWAT. Hence, the computational costs prevents the use of a fine grid (e.g. Sreekanth and Datta, 2010).*
*Furthermore, according to Guo et al., (2002) experience suggests that 10 model layers per aquifer unit seem to be adequate, but users are encouraged to perform numerical experiments with different levels of grid resolution in order to determine the most appropriate number of layers. Experience also has shown that models designed with spatially uniform cell volumes are less prone to numerical instabilities than models designed with variable cell volumes.*

*Then, following these recommendations we have used spatially uniform cell volumes and 11 model layers.*

*-Guo, Weixing, and Langevin, C.D., 2002, User's guide to SEAWAT: A computer program for simulation of three-dimensional variable-density ground-water flow: U.S. Geological Survey Techniques of Water-Resources Investigations, book 6, chap. A7, 77 p.*

*-Sreekanth J, Datta B. 2010. Multi-objective management of saltwater intrusion in coastal aquifers using genetic programming and modular neural network based surrogate models. Journal of Hydrology 393: 245–256.*

**9) It is stated that inverse modeling is not used "due to the complexity of the case study dealt with". Does that imply that auto-calibration cannot be used for complex systems? If that is what you mean, please argue why.**

*The reviewer is right. We apologize for not expressing ourselves sufficiently clearly. We have not used an inverse model because is kind of cumbersome to deal with such quantity of parameters (hydraulic conductivity, storativity, porosity, dispersion coefficients ...) and variables (hydraulic head and salt concentrations) over a long period of time (from 1973 to 2010), so that we have opted to apply a trial and error procedure.*

*We have rewritten this phrase to avoid misunderstanding and possible interpretations of the reader regard that auto-calibration cannot be used for complex systems.*

**10) As a minimum the match to the observations should be quantified by a few statistics (e.g., Mean Error, Root Mean Squared value)**

*In accordance with the reviewer's comment we will add a few statistics. The root mean square value (RMS) of the departures between observed and simulated values for both piezometric heads and salt concentrations is presented for the whole domain and temporal discretization due to the large number of boreholes (21 boreholes for piezometric heads and 31 for salt concentrations). These values are:*

$$\eta_h = 0.7 \ m \qquad\qquad ; \ \eta_c = 191.8 \ mg/l$$

*Note that this $\eta_h$=0.7 m could seems to be a little high but we should take into account that we have observation wells where the historical hydraulic head measurement fluctuates sometimes more than 6m during the same month (see for example observation well 6).*

*Note that the high value for $\eta_c$ =191.78 mg/l could be also explained by the scale of the concentration, which range from 0 to 35000 mg/l, and the measurement fluctuations, which in some wells is even higher than 4000 mg/l during some months.*

*It would be desirable to have better adjustments between observed and modeled hydraulic head and salinity to have a greater confidence in the predictions made with the model. However, it is difficult to improve the approach proposed in this work for Plana de Oropesa-Torreblanca due to the following facts:*

*- The hydrogeological complexity of the aquifer makes it difficult to define a better approach taking into accunt its spatial heterogeneity, with fracturated formations and preferential flow channels existing in the aquifer (Morell and Giménez, 1997). The spatial heterogeneity is handled by means of sequential indicator simulation using the computer code ISIM3D (Gómez-Hernández and Srivastava, 1990). See more details about how this heterogeneity is handle in the answer to the comments 5 and 7 of reviewer 2.*

*- The quality of some observation data is not very good. This problem is accentuated when considering the long simulated time period with historical data, which spans from 1973 to 2010. In this way, for a certain observation borehole there are data close in time with measurements quite disparate, which cannot be explained by any physical phenomenon. A statistical processing of data with expert judgment would be advisable to dismiss the wrong data. We have opted for using all available data (as a transparency measurement and in order to ease the reproduction of this exercise by other researchers).*

*- The lack of reliable estimates of dispersion coefficients (Naji et al., 1999) may prevent a proper adjustment for the Plana Oropesa-Torreblanca aquifer.*

*- A better fit might be achieved using a more refined spatial discretization, which would allow modeling the preferential flow channels existing in the aquifer (Morell and Giménez, 1997). However, the high computational burden when solving the variable-density flow and transport equations with SEAWAT prevents the use of a fine grid (e.g. Sreekanth and Datta, 2010).*

*- As a further research, we are intended to couple the salt water intrusion model with a management simulation–optimization model for control and remediation that would further prevent the use of a fine grid.*

*As a conclusion, we have opted for using a variable-density model instead of sharp-interface solutions (most of them based on the Ghyben–Herzberg relation), which have been extensively employed to define management models because of its simplicity in terms of required parameters and computational burden (e.g., Mantoglou et al., 2004). This is because they better describe the dynamic of real complex coastal aquifers despite the limitations in the available data and the course discretization used for the Plana Oropesa-Torreblanca aquifer. These drawbacks lead to some differences between observed and simulated data, especially in the salinity concentration. Nevertheless, the final model seems to be good enough to capture the general trend and to assess the impacts of climate and land-use and land-cover changes, which is the main aim of the present work. Instead of using a sharp-interface solution that does not take into account the diffusion and dispersion mechanisms we propose a more physical approach to approximate the dynamic of the salinity concentration.*

*In order to clarify these issues within the new version of the manuscript we intend to add a **limitation section,** in which all these issues and considerations would be added and highlighted.*

*Nevertheless, although due to the differences between observed and simulated data the uncertainty of the prediction coming from the model grows and we have to be be cautious with the conclusions obtained in the application to the case study (aspect included in the limitation section), we think that the fit is good enough to capture the general trend of the hydraulic head and salinity variables within a quite long calibration period (37 years, from 1973 to 2010), and therefore, to assess general impacts of climate and land-use and land-cover changes. On the other hand, from a methodological point of view, as the reviewer 1 pointed in his first comments, the proposed approach is ambitious and valid scientifically, which is the main scope of this research work.*

**11) REFERENCES:**

*\*Mantoglou A, Papantoniou M, Giannoulopoulos P. 2004. Management of coastal aquifers based on nonlinear optimization and evolutionary algorithms. Journal of Hydrology 297(1-4): 209 228. DOI: 10.1016/j.jhydrol.2004.04.011*

*\*Morell, I. and Giménez, E.: Hydrogeochemical analysis of salinization processes in the coastal aquifer of Oropesa (Castellón, Spain), Environmental Geology, 29(1/2), 118-131, 1997.*

*Naji A, Cheng AD, Quazar D. 1999. BEM solution of stochastic seawater intrusion problems. Engineering Analysis with Boundary Elements 23: 529–37.*

*Sreekanth J, Datta B. 2010. Multi-objective management of saltwater intrusion in coastal aquifers using genetic programming and modular neural network based surrogate models. Journal of Hydrology 393: 245–256.*

**12) Future climate signals are found by averaging the results from the available climate models and subsequently feed this averaged signal into the hydrological models. Alternatively, results from each individual climate model should have been used as input to the hydrological model system and averaged afterwards. Please document that the method used is appropriate.**

*The reviewer is right. In order to assess the uncertainty on hydrological impacts it would be more appropriate to obtain results from each individual climate model. Nevertheless, in this paper we do not intend to perform a detailed analysis of uncertainty, which could be deeply analyzed as a further research. Our main target is to provide an estimate of the most representative plausible future climate scenarios. For this reason we propose to simulate 4 plausible representative climate scenarios defined by ensemble of different climate models, which provide a better approach of future climate scenarios than taking directly a scenarios defined by a single model. The 'ensembles' coalesce and consolidate the results of individual climate projections, thus allowing for more robust climate projections that are more representative than those based* on a single model (Spanish Meteorologial Agency, AEMET, 2009).

*We think that we made a mistake including the word uncertainty in the title and we propose to remove it, because it could produce misunderstand about the target of the paper.*

*REFERENCE:*
AEMET, 2009. Generación de escenarios regionalizados de cambio climático para España. Agencia Estatal de Meteorología, Mto. Medio Ambiente, Medio Rural y Marino.

**13) Details on the downscaling methods completely missing. There are many versions of what you call "bias correction" – which one did you use?**

*We used a correction of the first and second moments analogous to those applied by Pulido-Velazquez et al. (2014) for the delta change approach. The difference in the bias correction approach the perturbation is calibrated by modifying some statistics (first and second moments) of the control series in order to make them identical to the historical ones. It assumes that this perturbation will be maintained invariant during the future.*

*REFERENCE:*

*Pulido-Velazquez, D., García-Aróstegui, J.L., Molina, J.L., and Pulido-Velázquez, M.: Assessment of future groundwater recharge in semi-arid regions under climate change scenarios (Serral-Salinas aquifer, SE Spain). Could increased rainfall variability increase the recharge rate?, Hydrol. Process., 29(6), 828-844, doi:10.1002/hyp.10191, 2014.*

**14) How was the delta change method applied – monthly, yearly?**

*It was applied monthly in analogous way as presented in a previous work by Pulido-Velazquez et al. (2014)*

*REFERENCE:*
*Pulido-Velazquez, D., García-Aróstegui, J.L., Molina, J.L., and Pulido-Velázquez, M.: Assessment of future groundwater recharge in semi-arid regions under climate change scenarios (Serral-Salinas aquifer, SE Spain). Could increased rainfall variability increase the recharge rate?, Hydrol. Process., 29(6), 828-844, doi:10.1002/hyp.10191, 2014.*

**15) RESULTS: The result section is very short and does actually not explain why the presented results are obtained. For example, why is the impact of sea level rise to insignificant? What is most important – climate change or LULC changes?**

*We agree with the reviewer. Following his suggestion we will extend the results section in order to explain why the presented results are obtained:*

The proposed approach allows to assess the impacts of different climatic and land uses change scenarios in terms of global flow balance, as well as a distributed approximation of the hydraulic head and salinity. The components of the global balance for the simulated future scenarios show that, in general terms, the intrusion problems will not grow and will even be reduced slightly, as the outflows to the sea will not decrease, due in part to the reduction and redistribution of pumping in the mentioned scenarios (see Figure 13). From the results in terms of salinity at specific observation points, we can identify areas where the situation will clearly improve throughout the future horizon contemplated with the proposed scenarios. For example, at observation point 21 (in the southern area) the salinity would be reduced with the contemplated scenarios. In other areas, around the observation point 12, the situation deteriorates significantly until arriving at the last years of the horizon 2035. As commented above the expected results without considering global change impacts are likely to be too pessimistic or optimistic, depending on the location. These results can be useful to the authorities in charge of the planning and definition of management policies in the Plana de Oropesa Torrablanca. Modifying the input of the the integrated modeling framework developed it could be useful to assess potential effect of adaptation measures to global change. Participatory processes including the relevant stakeholders might be essential in the successful definition of adaptation measures for

groundwater management (Pulido-Velazquez et al., 2015). This modeling framework could be useful in the search for consensus - "shared vision" models.

*For example, we will answer the questions asked by the reviewer:*

The maximum value of SLR considered, 0.19m in 2035, is quite low with respect to the level fluctuations experienced in most of the observation wells (see Figure 15). For this reason the sensitivity of the flow and transport is low.

We have also defined an additional scenario and we are currently simulating it. It considers future LULC assuming that there is not climate change, which would help to analyses and discuss in a quantitative way the relative significance of climate change and LULC in the final impacts.

**16) DISCUSSION: - There is no discussion of the results and this is critical. The manuscript cannot be published without a proper discussion of the results. This includes a comparison of methods and with results from other studies.**

*Following the reviewers' comment we will do our best to improve the discussion section within the revised version of the manuscript. As commented in the answer to the previous reviewer question we will improve the result section, including and explaining new results. On the other hand we will also discuss the methods and results comparing with other previous approaches and studies. In this answer we include examples of some of the changes that would be introduced in the future manuscript.*

*For example, from a methodological point of view, the proposed approach has some similarities with that proposed by Pulido-Velazquez et al. (2015), in which an integrated analysis of global change is performed including climate and LULC changes. The most important difference is that in this case a coastal aquifer is studied and a variable density model is used to propagate the impacts on it.*

*For example, with respect to the sensitivity of the SLR on the results, we find in the literature other examples in which the sensitivity of seawater intrusion to the SLR would be low. Chan et al. (2011) obtained this conclusion in a synthetic confined coastal aquifer in which recharge in unchanged; Rasmussen et al. (2013) obtained the same conclusion for an inland coastal aquifer with minor SLRs. In our case the maximum value of SLR considered is 0.19m (in 2035), and it is quite low with respect to the level fluctuations experienced in most of the observation wells (see Figure 15). For this reason the sensitivity of the flow and transport is low. Nevertheless other authors, as Werner and Simmons (2009) showed that in unconfined aquifers the influence of the inland boundary condition can be significant to its sensitivity to SLR.*

*Finally we will also summarize pros and cons of our study within a new subsection included in the discussion, a "limitation of the results and future research works" subsection. For example, in the **limitation** we would explain why it is difficult to improve the approach proposed for Plana de Oropesa-Torreblanca although it would be desirable to have better adjustments between observed and modeled hydraulic head and salinity to have a greater confidence in the model predictions (see response to question 10). Nevertheless, although due to the differences between observed and simulated data the uncertainty of the prediction coming from the model grows and we have to be cautious with the conclusions obtained in the application to the case study (aspect included in the limitation section), we think that the fit is good enough to capture the general trend of the hydraulic head and salinity variables within a quite long calibration period (37 years, from 1973 to 2010), and therefore, to assess general impacts of climate and land-use and land-cover changes. On the other hand, from a methodological point of view, as the reviewer 1 pointed in his first comments, the proposed approach is ambitious and valid scientifically, which is the main scope of this research work.*

*As we show in the answer to the next reviewer comment (comment number 16), although the assessment of uncertainty out of the scope of the present paper, a proper analysis of it could be performed in **future research works**.*

**17) Uncertainty: The uncertainty of the results are not touched at all. Considering the chain of model component that are used the total uncertainty of the obtained results must be significant. A discussion of this element is mandatory. Quantification would be even better.**

*We think that we made a mistake including the word uncertainty in the title and we propose to remove it, because it could produce misunderstand about the target of the paper. We agree with the reviewer that a deeper and broader treatment of the uncertainty would be advisable. However, we consider this is out of the scope of the present paper. Note that it would require to deal with different sources of uncertainty. The complexity is even greater for the presented methodology, since it entails the coupling of several numerical codes and a large amount of data and a long simulation time period.*

*There are numerous classification schemes for sources of uncertainty in the literature. In this sense, the uncertainties covered in this work could be summarized as (Matott et al., 2009):*
*-Parameter, model, and modeller uncertainty.*
*-Initial system state, parameter, input, and output uncertainty.*
*-Context, input, parameter, structural, and technical uncertainty.*
*-Statistical variation, subjective judgment, linguistic imprecision, variability, inherent randomness, disagreement, approximation.*

*There are also a lot of quantitative methods and tools for uncertainty assessment in integrated models (Matott et al., 2009), which would be worth a paper by itself when applied to the Plana Oropesa-Torreblanca aquifer.*

*Data analysis (DA): to evaluate or summarize input, response, or model output data.*

*Identifiability analysis (IA): to expose inadequacies in the data or suggest improvements in the model structure.*

*Parameter estimation (PE): to quantify uncertain model parameters using model simulations and available response data.*

*Uncertainty analysis (UA): to quantify output uncertainty by propagating sources of uncertainty through the model.*

*Sensitivity analysis (SA): to determine which inputs are most significant screening, local, global.*

*Multimodel analysis (MMA): to evaluate model uncertainty or generate ensemble predictions.*

*Bayesian networks (BN): to combine prior distributions of uncertainty with general knowledge and site-specific data to yield an updated (posterior) set of distributions.*

*As a further research we could apply some of these models and techniques to deal with the uncertainty.*

*Following the reviewers' comment these considerations have been added to the manuscript.*

*- Matott, L. S., J. E. Babendreier, and S. T. Purucker (2009), Evaluating uncertainty in integrated environmental models: A review of concepts and tools, Water Resour. Res., 45, W06421, doi:10.1029/2008WR007301.*

---

## Author Response (AR1)

**Journal: Hydrol. Earth Syst. Sci.**
**Manuscript Number:  hess-2017-262**
**Title: "Integrated assessment of future potential global change scenarios and their hydrological impacts in coastal aquifers. A new tool to analyse management alternatives in the Plana Oropesa-Torreblanca aquifer"**
**Special Issue: Assessing impacts and adaptation to global change in water resource systems depending on natural storage from groundwater and/or snowpacks**

**\*\*\*\*\*\*\*\*\*\*\*\*\*\*\*\*\*\*\*\*\*\*\*\*\*\*\*\*\*\*\*\*\*\*\*\*\*\*\*\*\*\*\*\*\*\*\*\*\*\*\*\*\*\*\*\*\*\*\*\*\*\*\*\*\*\***

**Comments from the editors:**

**\*\*\*\*\*\*\*\*\*\*\*\*\*\*\*\*\*\*\*\*\*\*\*\*\*\*\*\*\*\*\*\*\*\*\*\*\*\*\*\*\*\*\*\*\*\*\*\*\*\*\*\*\*\*\*\*\*\*\*\*\*\*\*\*\*\***

**Comments to the Author:**
**The manuscript requires major revision and additional review before publication. Please consider the comments of the reviewers carefully before submitting a new improved version of the manuscript. Especially reviewer#2 raise some serious issues that need to be carefully dealt with. The new version of the manuscript should be accompanied by detailed descriptions of how the new version considered the critical points raised by the reviewers**

**Response to Comments from the Editors:**
*We appreciate the Editor for giving us the opportunity to provide a new version of the manuscript. We have made an important effort to improve the manuscript in accordance with the editor and reviewer comments.*
* * *
**Reviewer #1 Comments to Author:**
* * *
1) **This paper defines a framework for assessing the effects of CC, LULC and SLR on a coastal aquifer where problems of saltwater intrusion are detected. Climate change models are integrated with LULC, SLR and double-density groundwater flow models in order to define future strategies for integrated water management in the study area. The approach is ambitious and valid scientifically. Many models, however, are introduced but not clearly explained. Some models are described with excessive jargon and others are barely defined and no reference is given (modified etr). As a result, the paper has a black box kind of content, which makes it very difficult to evaluate. I suggest that the authors upload some additional information that relates to the different models and steps they did in their work.**

*We thank the reviewer for the recognition of the interest of this research and for the comments formulated, which have helped us to realize that some aspects were not clearly stated in the original manuscript and could be improved in a new version of the manuscript. Following the reviewer suggestion we will upload some additional information that relates to the different models and steps we did in our work. More detail about it can be seen in the response to the specific comments.*

*For example, see the response to comment number 6, in which a more detailed description of the rainfall recharge model is proposed in order to understand and evaluate it.*

2) **Another confusing point is the use of the acronym GC, GCi, GC1, GC2, etc. which are never correctly explained.**

*Following the reviewer suggestion we have defined each of the employed acronyms within the new version of the manuscript.*

3) **The authors use also inconsistencies in defining concentration and relating it to density (density 1025 Kg/m3; salinity 1035 g/l), which can cause serious problems in the double-density models. It is not clear if in the models they use chloride concentration or salinity.**

*The salinity concentration of the seawater that we have used is 35 g/l in the SEAWAT simulations. We have used this value throughout all the revised version of the manuscript.*

4) **Also the porosity used seems very small compared to the permeability detected in the aquifer.**

*Sorry, there was a typo error in this paragraph. The specific yield takes values from 0.01 to 0.15 and effective porosity values from 0.01 to 0.13; We have already corrected it within the new version of the manuscript (line 15-19 page 10):*

The calibrated model parameters encompass different zones of horizontal hydraulic conductivities ranging from 5 to 200 m/d, while the vertical hydraulic conductivities are between 0.5 to 20 m/d. There are also different zones of specific storage values, which range from 10-5 to 5·10-4 1/m; specific yield, with values from 0.01 to 0.15; effective porosity values from 0.01 to 0.13; and dispersion coefficients from 50 to 100 m.

*Nevertheless, in in order to explain more clearly the conceptual approach followed to define the model parameters we have added some sentences explaining it within the new version of the manuscript (line 18-22 page 10):*

We have replaced a highly heterogeneous porous media with an upscaled "equivalent" homogeneous porous media to represent the hydrogeological parameters since the cell size of the discretization is 250x250 m (e.g., Llopis-Albert and Capilla, 2010). Then, we have used a value of the effective porosity based on available data, which were subsequently calibrated and upscaled using expert judgment. The results of the calibration process prove the worth of this approach.

*-Llopis-Albert, C., Capilla, J.E. (2010). Stochastic inverse modelling of hydraulic conductivity fields taking into account independent stochastic structures: A 3D case study. Journal of Hydrology, 391(3–4), 277-288. DOI: 10.1016/j.jhydrol.2010.07.028.*

**5) Another point that the authors do not address is the connection between the carbonate basement and the detritic aquifer.**

*Thank to the reviewer comment we have realized that we did not explain it properly within the manuscript. In order to clarify it we have added the next sentences (Section 2.1; line 33-34 page 3):*

The carbonate basement receives the contributions coming from the bordering aquifer (Maestrazgo aquifer) and feeds the detrital aquifer. The direction of the groundwater flow is from the carbonate basement to the detrital aquifer.

*We have also added the next sentence in section 3.2.1 (rainfall-recharge models):*

The Maestrazgo aquifer recharge model is used to assess inflows to the carbonate basement, that produces lateral inflows (LI) to the Plana Oropesa-Torreblanca aquifer under various potential GC scenarios. The Maestrazgo aquifer has an important storage capacity which is almost in natural regime and does not present significant changes in hydraulic head. The Maestrazgo rainfall recharge model is employed to assess future recharge being the LI to the carbonate basement obtained by assuming a constant ratio between Maestrazgo rainfall recharge and these LI.

**6) Section 3.1.1 This section requires better explanation of the modeling (lines 20-30).**

*In the next paragraphs we include a more detailed description of the rainfall-recharge model:*

Based on the historical climate (rainfall and T) and recharge series for the period 1973-2010 described in sections 2.2, we define a simple empirical rainfall-recharge approach to generate yearly aquifer recharge series. The model assumes that P and T are the most important climatic variables determining potential aquifer recharge, and the variability of both of them will determine the impacts of future potential climatic scenarios.

It intend to define a correction function for the perturbation of the historical series defined as the difference in P and evapotranspiration (ETR) (hereafter referred to as PE series), modifying its mean and standard deviation to make them equal to the statistic of the historical aquifer recharge previously deduced from the lysimeter measurements (see section 2.2), as follows:

$$R_i = PEn_i . \sigma_R + \bar{R} \qquad (1)$$

where $R_i$ is the recharge series generated for the year i, $\sigma_R$ and $\bar{R}$ are the standard deviation and mean of the historical recharge series estimated using the infiltration rate coefficient obtained from previous lysimiter readings (Tuñon, 2000), and $Pn_i$ is the normalised historical PE series (P-E) obtained from:

$$Pn_i = \frac{PE_i - \overline{PE}}{\sigma_{PE}} \qquad (2)$$

where $PE_i$ is the historical PE series for the year i, and $\sigma_{PE}$ and $\overline{PE}$ are the standard deviation and mean historical values of the series. Taking the positive relationship of T and E into account (Arora, 2002; Gerrits et al., 2009), changes in T will determine the available non-evaporative fraction of P available for aquifer recharge. Different non-global empirical models could be applied to assess the historical E from T series (e.g., Turc, 1954, 1961; Coutagne, 1954; Budyko, 1974; amongst others) as described in Arora (2002), Gerrits et al. (2009), and España et al. (2013). In this study, we applied Turc's model (1954, 1961), in which the results depend on mean annual T and solar irradiation over the latitude.

In order to approach the impact of seasonal variability the annual rainfall recharge values obtained using the simplified model were distributed between the 12 months maintaining the pattern of the historical rainfall recharge series. We assume that recharge from a rainfall event will reach the aquifer in less than one month, so working with stress periods of one month means there is no delay between the rainfall and the aquifer recharge.

NEW REFERENCES:
*Arora, V.K.: The use of the aridity index to assess climate change effect on annual runoff, Journal of Hydrology, 265, 164-177, 2002.*
*Budyko, M.I.: Climate and Life, Academic Press, New York, 508pp, 1974.*
*Coutagne, A.: 1954. Quelques considérations sur le pouvoir évaporant de l'atmosphere, le déficit d'écoulement effectif et le déficit d'écoulement maximum, La Houille Blanche, 360-369, 1954.*

*España, S., Alcalá, F.J., Vallejos, A., and Pulido-Bosch, A.: A GIS tool for modelling annual diffuse infiltration on a plot scale, Computers & Geosciences, 54, 318-325, 2013.*

*Gerrits, A.M.J., Savenije, H.H.G., Veling, E.J.M., Pfister, L.: Analytical derivation of the Budyko curve based on rainfall characteristics and a simple evaporation model. Water Resour. Res., 45, W04403, doi:10.1029/2008WR007308, 2009.*

*Kirn, L., Mudarra, M., Marín, A., Andreo, B., and Hartmann, A.: Improved Assessment of Groundwater Recharge in a Mediterranean Karst Region: Andalusia, Spain. In: Renard P., Bertrand C. (Eds.) EuroKarst 2016, Neuchâtel, Advances in Karst Science, 117-125, Springer, Cham, DOI: 10.1007/978-3-319-45465-8_13, 2017*

*Turc, L.: Water balance of soils: relationship between precipitation, evapotranspiration and runoff (in French), Ann Agron 5, 49-595 and 6, 5-131, 1954.*

*Turc, L.: Estimation of irrigation water requirements, potential evapotranspiration: A simple climatic formula evolved up to date, Annales Agronomiques, 12(1), 13-49, 1961.*

**7) I find the correlation between observed and modeled hydraulic head and salinity very poor. Can the authors explain on which basis the results of the models are acceptable as a predictive tool?**

*Thanks to the reviewer comment we have realized that we did not explain it properly within the manuscript. In order to highlight and clarify these issues within the new version of the manuscript we have added some paragraphs in a new **section** (Section 4.1 Hypotheses and limitations. Usefulness of the results). We agree with the reviewer, as he pointed in his first comments, from a methodological point of view the proposed approach is ambitious and valid scientifically, but we need to clarify assumptions and limitations in the application performed to the case study. We have added the next paragraphs (line 20 page 13 – 6 page 14)*

In this study we decided to employ a variable-density model instead of a sharp-interface solution, which have been extensively used to define management models because of its simplicity in terms of required parameters and computational burden (e.g., Mantoglou et al., 2004). This is because they better describe the dynamic of real complex coastal aquifers despite the limitations in the available data and the course discretization used for the Plana Oropesa-Torreblanca aquifer. Better adjustments between observed and modeled hydraulic head and salinity would provide greater confidence in the model predictions. Nevertheless, although due to there are some differences between observed and simulated data the uncertainty of the prediction coming from the model grows and we have to be cautious with the conclusions obtained, the fit is good enough to capture the general trend of the hydraulic head and salinity variables within a quite long calibration period (37 years, from 1973 to 2010), and therefore, to assess general impacts of climate and LULC changes. Note that it is difficult to improve the calibration of the proposed approach in this work for Plana de Oropesa-Torreblanca due to the following facts:

- The hydrogeological complexity of the aquifer makes it difficult to define a better approach taking into account its spatial heterogeneity, with fractured formations and preferential flow channels existing in the aquifer (Morell

and Giménez, 1997). The spatial heterogeneity is handled by means of sequential indicator simulation using the computer code ISIM3D (Gómez-Hernández and Srivastava, 1990).

- The quality of some observation data is poor. This problem is accentuated when considering the long simulated time period with historical data, which spans from 1973 to 2010. In this way, for a certain observation borehole there are data close in time with measurements quite disparate, which cannot be explained by any physical phenomenon. A statistical processing of data with expert judgment would be advisable to dismiss the wrong data. We have opted for using all available data (as a transparency measurement and in order to ease the reproduction of this exercise by other researchers).

- The lack of reliable estimates of dispersion coefficients (Naji et al., 1999) may prevent a proper adjustment for the Plana Oropesa-Torreblanca aquifer.

- A better fit might be achieved using a more refined spatial discretization, which would allow modeling the preferential flow channels existing in the aquifer (Morell and Giménez, 1997). However, the high computational burden when solving the variable-density flow and transport equations with SEAWAT prevents the use of a fine grid (e.g. Sreekanth and Datta, 2010).

- As a further research, we are intended to couple the sea water intrusion model with a management simulation–optimization model for control and remediation that would further prevent the use of a fine grid.

REFERENCES:

*Mantoglou A, Papantoniou M, Giannoulopoulos P. 2004. Management of coastal aquifers based on nonlinear optimization and evolutionary algorithms. Journal of Hydrology 297(1-4): 209 228. DOI: 10.1016/j.jhydrol.2004.04.011

*Morell, I. and Giménez, E.: Hydrogeochemical analysis of salinization processes in the coastal aquifer of Oropesa (Castellón, Spain), Environmental Geology, 29(1/2), 118-131, 1997.

*Naji A, Cheng AD, Quazar D. 1999. BEM solution of stochastic seawater intrusion problems. Engineering Analysis with Boundary Elements 23: 529–37.

*Sreekanth J, Datta B. 2010. Multi-objective management of saltwater intrusion in coastal aquifers using genetic programming and modular neural network based surrogate models. Journal of Hydrology 393: 245–256.

8) The discussion and conclusion sections are very short and poorly quantitative and fail to point out how this kind of modeling can be used in integrated coastal water management. The authors should elaborate on their results and say explicitly how this knowledge can be used in an integrated water management framework of a coastal zone. Give also explicit examples of how this can be done.

Following the reviewers' comment we have done our best to improve the DISCUSSION SECTION within the revised version of the manuscript.

[revised manuscript text omitted]

*As we show in the answer to the next reviewer comment (comment number 16), although the assessment of uncertainty out of the scope of the present paper, a proper analysis of it could be performed in **future research works**.*

*We have also rewritten the **CONCLUSION SECTION** in accordance with the discussion peformed:*

We have proposed a method to perform an integrated analysis of the potential impacts of future CC, LULC change and SLR in a coastal aquifer. It has been applied in the Plana Oropesa-Torreblanca aquifer assuming some hypotheses or simplifications.. Representative future CC scenarios are generated by using different equifeasible and non-equifeasible (deduced form a multicriteria analysis) ensemble of series generated with several RCMs applying different downscaling approaches (bias or delta change corrections). A future LULC scenario was defined in accordance with the plan approved by the local government (PGOU, 2009). Four GC scenarios were defined by

combining the LULC scenario and the CC scenarios. These GC scenarios have been propagated to assess hydrological impact by simulating them within a coupled modelling framework based on density-dependent model whose inputs are defined by a sequential coupling of different models (rainfall-recharge models, crop irrigations requirements and irrigation return models). These global scenarios' simulations show a significant increase (respect to a BL scenario with no GC) in the variability of the flow budget components and in the salinity. In global terms the intrusion will not grow in the considered potential future scenarios. The impacts on the aquifer salinity will be heterogeneous. We observed specific areas where the situation gets worse and other, where it will clearly improve in the contemplated future horizon due to transformation of irrigated areas to residential use foreseen in the future LULC scenario. They also show a low sensitivity to an extreme SLR scenario, especially in terms of hydraulic head. The proposed analysis is valuable to improve our knowledge about the aquifer and so comprise a tool to support decisions about sustainable adaptation management strategies. We can use this coupled modeling framework to assess potential adaptation measures under different GC scenarios and horizons changing the inputs of the models. In the definition of scenarios and plausible adaptation measures to be analyzed participatory processes including the relevant stakeholders might be essential (Pulido-Velazquez et al., 2015). On the other hand, this modeling framework could be useful in the search for consensus ("shared vision" models) between different stakeholders.

**9) SOME SPECIFIC POINTS ABOUT THE FIGURES:**

**Figure 1: Vertical scale is missing in the figure. Not discussed in the text is the relationship between the carbonate rocks and the detritic aquifer. No explanation of the lithotype in the geologic time scale legend is given. There are too many eastings and northings in the map. Define them only at the corners of the figure. Confusing the color grey used for the aquifer and the Mediterrenean Sea.**

*Following the reviewer suggestion we have updated Figure 1.*

[Figure]

**Figure 2: The CORINE database is not mentioned in the text.**

*Following the reviewer suggestion we have mentioned it within the new version of the manuscript (line 8-10 page 4):*

1) Changes in LULC (Figure 2), obtained from both fieldwork undertaken in the area and from the European CORINE Land Cover database (Feranec et al., 2010). These data were used to estimate the irrigation returns, following the procedure described in Section 3.2.2.

**Figure 3: The overlap does not allow to distinguish well the data from the two watersheds. Also the choice of color is poor. Maybe use the same color for the same watershed.**

*We have eliminated the overlap in the monthly data and we have changed other aspects to clarify the figure.*

[Figure]

**Figure 5: Please give also some information about the fact that you are presenting climate models data. This caption is not sufficient to understand what kind of data are presented.**

*Done, we have modified the figure caption.*

Figure 6. Monthly mean and standard deviation of the historical and RCMs control series (rainfall and temperature) for the mean year in the period 1976-2000. RCMs data obtained from CORDEX project.

**Figure 6: See my note above. Also here some more information is needed. At least give the time frame for the climate change models.**

*Done, we have modified the figure caption.*

Figure 7. Relative monthly change in mean and standard deviation of the future series (2011-2035) with respect to the control series (1976-2000) for the considered RCMs under the RCP8.5 emission scenario.

**Figure 7: I would have presented this figure much earlier on in the paper.**

*Following the reviewer suggestion we will include it earlier in the new version of the manuscript (Figure 4 instead of Figure 7). This reviewer comment is also linked with the comment number 2 of the reviewer 2 about the organization of the manuscript (Chapter 2.3, 2.4 and 2.5 should be moved to Methods)*

**Figure 8: In wells 6, 23, 20, 8, and 21 there is a large difference between observed and modeled hydraulic head data. This, in a coastal context is not a good thing, because it makes the results of the double-density flow model unreliable. I think that the authors should address this large variability and explain how their flow model is still acceptable in view of this poor correlation.**

*See the answer to comment number 7*

**Figure 8: I find the correlation between observed and modeled salinity very poor also here. Can the authors explain on which basis the results of the models are acceptable as a predictive tool.**

*See the answer to comment number 7*

**Figure 9: It would be nice to separate the inflow from the outflow in this graph, so that it is clear the variation in the total yearly budget (you can do this by using the same color for inflows and different data point symbols; and a different color for outflows . with different data symbols).**

*Following the reviewers' comment we have separated the inflows (orange) and the outflows (blue) in Fig. 10 (Figure 9 in the previous version) and 12.*

[Figure]

**Fig. 9**

**Figure 11: Specify data are at monthly level.**

*Done, we have provided the information in the figure caption.*

Figure 9. Monthly mean and standard deviation of future precipitation and temperature series obtained by the four ensemble options.

**Figure 12: See my note for Figure 9.**

*As in Figure 10, we have separated the inflows and outflows in the graph.*

[Figure]

[Figure]

**Fig. 12**

**Figure 13: x axis should be "water budget components". Please specify a little bit better what the different CG's are. Hm3 / year is not a standard flow unit. Please specify.**

*Sorry for the mistake. It is a typo error. It should be GC scenarios (global change scenarios) instead of CG. We have corrected it. The units are Millions of cubic meters per year (Mm3/year), we will also correct it in the new version.*

[Figure]

**Figure 13**

**Figure 14: A few words about well locations in the text would be helpful. Also, sometimes you talk about salinity and sometimes about chloride concentration. They are not the same thing. Could you please explain in the text what concentrations unit you are using and why?**

*Following the reviewer suggestion we will include some words describing the location of the wells represented in Figure 14.*

Figure 14 shows the evolution in terms of salinity at 4 specific observation points roughly equispaced. They were selected to cover the extension of the aquifer from north to south (starting with the more northerly and moving towards the south we have observation wells 33, 12, 39 and 21 respectively).

From the results in terms of salinity at specific observation points (Figure 14), we can observe the heterogeneity of the impacts of the LULC scenario and CC scenarios. The area around the observation point 33 is not affected by LULC changes and we only observe sensitivity to CC scenarios that produce a higher variability in the salinity evolution, but the mean trend of the concentration does not change. Around the observation point 12, in Torreblanca area, the LULC change scenario would produce a reduction in the recharge whose impacts can be observed as an

increment in the salinity during the first decade of the future horizon. Nevertheless, in the last 10 years the reduction in pumping produced by the successive transformations of irrigated areas to residential land would reduce significantly the salinity. In this Torreblanca area CC scenarios would impact the salt concentration significantly during the first years due to the increment in pumping requirements produced by the higher water irrigation requirements obtained for these CC scenarios. Even considering the impacts of CC scenarios, during the last 10 year of the horizon it starts to recover due to the reduction in pumping produced by the new transformations of irrigated areas to residential land defined in the LULC scenario, being the concentration at the end of the horizon (2035) even under the values obtained for the BL scenario. In the observation point 39, located further away from the coast, the reduction in pumping due to LULC change would reduce the salinity during most of the year of the horizon. Nevertheless the reduction in recharge makes in some years the salinity to get close to the one obtained without LULC changes (BL scenario), being very similar at the end of the period. Again CC scenarios would increase the variability of the salinity in the simulated period.

We can also identify in the southern part areas where the situation will clearly improve throughout the future horizon contemplated with the proposed scenarios. For example, at observation point 21, in Oropesa area, the salinity would be reduced with the contemplated scenarios, which would be mainly related with the reduction of pumping in this area due to the transformation of irrigated areas to residential land defined in the LULC scenario.

*On the other hand, as the reviewer pointed out we have used both terms indistinctly in the text and we have made a mistake when using it within the Figure 14 caption. Instead of chloride concentration it should be salinity concentration. We have used both terms within the manuscript because data is provided as chloride concentration, while in SEAWAT simulations we use salinity (mg/l) as concentration unit. The conversion is performed according to the following equation (e.g., Williams and Sherwood, 1994):*

$$S \left(^o/_{oo}\right) = 1.80655 \times Cl^- \left(^o/_{oo}\right)$$

*where S is salinity and Cl- is Chlorinity.*

*Williams, W.D., Sherwood, J.E.(1994). Definition and measurement of salinity in salt lakes. International Journal of Salt Lake Research 3(1), 53–63.*

**10) I have attached a file with detailed requests for explanation in the text, some English corrections and suggestions. I hope this is helpful.**

*We sincerely appreciate the annotations provided by the reviewer in the attached file as complementary material. It has helped us to identify the paragraphs and sentences that are not clear enough. Following the reviewer suggestions we have modified and improved them within the new version of the manuscript. They have been really helpful to improve the clarity of the exposition.*

[Figure]

**Reviewer #2 Comments to Author:**
* * *
**1)** **Unfortunately, the manuscript is not ready for publication yet. Below, a number of critical issues are raised, including methods, discussion and results.**

*We have made an important effort in order to improve the manuscript in accordance with the valuable comments provided by both reviewers.*

**2)** **Organization: Chapter 2.3, 2.4 and 2.5 should be moved to Methods**

*Following the reviewers' comment we have moved those chapters to the Method section.*

**3) Methods: The applied modeling system is described as "integrated". However, there are no feed-backs in the system so it is misleading to call it integrated. A term like "coupled" would be more appropriate.**

*Following the reviewers' comment we have changed "integrated modeling framework" by "coupled modeling framework" throughout all the manuscript.*

**4) It is not clear how the rainfall-recharge model was calibrated – which data and which period. Results on calibration missing.**

*Thank to the reviewer comment we have realized that the rainfall recharge model needed a more detailed and clear description within the manuscript. In order to explain it more properly we have modified section 3.1.1 (rainfall Recharge model): See line Line 10-28 page 8:*

Based on the historical climate (rainfall and T) and recharge series for the period 1973-2010 described in sections 2.2, we define a simple empirical rainfall-recharge approach to generate yearly aquifer recharge series. The model assumes that P and T are the most important climatic variables determining potential aquifer recharge, and the variability of both of them will determine the impacts of future potential climatic scenarios.

It intend to define a correction function for the perturbation of the historical series defined as the difference in P and evapotranspiration (ETR) (hereafter referred to as PE series), modifying its mean and standard deviation to make them equal to the statistic of the historical aquifer recharge previously deduced from the lysimeter measurements (see section 2.2), as follows:

$$R_i = PEn_i . \sigma_R + \bar{R} \tag{1}$$

where $R_i$ is the recharge series generated for the year i, $\sigma_R$ and $\bar{R}$ are the standard deviation and mean of the historical recharge series estimated using the infiltration rate coefficient obtained from previous lysimiter readings (Tuñon, 2000), and $Pn_i$ is the normalised historical PE series (P-E) obtained from:

$$Pn_i = \frac{PE_i - \overline{PE}}{\sigma_{PE}} \tag{2}$$

where $PE_i$ is the historical PE series for the year i, and $\sigma_{PE}$ and $\overline{PE}$ are the standard deviation and mean historical values of the series. Taking the positive relationship of T and E into account (Arora, 2002; Gerrits et al., 2009), changes in T will determine the available non-evaporative fraction of P available for aquifer recharge. Different non-global empirical models could be applied to assess the historical E from T series (e.g., Turc, 1954, 1961; Coutagne, 1954; Budyko, 1974; amongst others) as described in Arora (2002), Gerrits et al. (2009), and España et al. (2013). In this study, we applied Turc's model (1954, 1961), in which the results depend on mean annual T and solar irradiation over the latitude.

*Next Figure shows the historical yearly evolution of the rainfall recharge in the aquifer obtained with the calibrated model and the historical series based on the lysimeter measurements:*

[Figure]

The low sensitivity of the results should be due to the maximum value of SLR considered, 0.19m in 2035, is quite low with respect to the level fluctuations experienced in most of the observation wells (see Figure 15). For this reason the sensitivity of the flow and transport solutions are low.

*In order to ask the question about the significance of climate change and LULC changes we have defined an additional scenario and we have simulated it. It considers future LULC assuming that there is not climate change, which would help to analyses and discuss in a quantitative way the relative significance of the impacts of climate change and LULC future scenarios.*

*The scenario has been defined in section 3.3:*

In order to assess the potential impacts of the future LULC scenario (LULC scenario) and different GC (CC and LULC change scenarios) we have simulated the following scenarios using the density-dependent flow model:

1) Baseline (BL) scenario: No LULC change and no CC. We simulate a future scenario for the horizon 2011-2035 assuming that from 2011 we would have the same LULC that we observed in 2010. We also assume that the hydrological characteristic does not change and we have simulated assuming the rainfall recharge and the LI from the neighbour aquifer are equal to those estimated in the last 5 years of the historical periods (2006-2010). In this period of 5 years (2006-2010) there was no significant change in LULC and so this period could be adopted as being representative of the mean recent climatic-hydrological conditions. This scenario was defined in order to compare against the others to analyse the sensitivity to GC.

2) LULC scenario: It considers the described future LULC scenario and assumes that there is not CC.

*The results obtained have been included in Figure 13 and 14.*

[Figure]

**Figure 13: Mean inflows and outflows for various global scenarios (GC1, GC2, GC3, GC4).**

[Figure]

**15) DISCUSSION: - There is no discussion of the results and this is critical. The manuscript cannot be published without a proper discussion of the results. This includes a comparison of methods and with results from other studies.**

*Following the reviewers' comment we have done our best to improve the results and discussion section (line 13 page 11 – 26 page 12) within the revised version of the manuscript. As commented in the answer to the previous reviewer question we have improved the result section, including and explaining new results. On the other hand we have also added some paragraph to discuss methods and results **comparing with other previous approaches** and studies.*

*For example, from a methodological point of view, we have added the next paragraph (line 14-16 page 11 of the new version of the manuscript) comparing with other previous studies:*

The proposed approach has similarities with the one described by Pulido-Velazquez et al. (2015), in which an integrated analysis of GC is performed including CC and LULC changes. The most important differences are related with the fact that a coastal aquifer is studied and we consider quality issues simulating with a variable density flow and transport model.

*In the discussion of results, the next comments have been added with respect to the significance of LULC with respect to CC in other previous studies:*

Figure 14 shows the evolution in terms of salinity at 4 specific observation points roughly equispaced. They were selected to cover the extension of the aquifer from north to south (starting with the more northerly and moving towards the south we have observation wells 33, 12, 39 and 21 respectively).

From the results in terms of salinity at specific observation points (Figure 14), we can observe the heterogeneity of the impacts of the LULC scenario and CC scenarios. The area around the observation point 33 is not affected by LULC changes and we only observe sensitivity to CC scenarios that produce a higher variability in the salinity evolution, but the mean trend of the concentration does not change. Around the observation point 12, in Torreblanca area, the LULC change scenario would produce a reduction in the recharge whose impacts can be observed as an increment in the salinity during the first decade of the future horizon. Nevertheless, in the last 10 years the reduction in pumping produced by the successive transformations of irrigated areas to residential land would reduce significantly the salinity. In this Torreblanca area CC scenarios would impact the salt concentration significantly during the first years due to the increment in pumping requirements produced by the higher water irrigation requirements obtained for these CC scenarios. Even considering the impacts of CC scenarios, during the last 10 year of the horizon it starts to recover due to the reduction in pumping produced by the new transformations of irrigated areas to residential land defined in the LULC scenario, being the concentration at the end of the horizon (2035) even under the values obtained for the BL scenario. In the observation point 39, located further away from the coast, the reduction in pumping due to LULC change would reduce the salinity during most of the year of the horizon. Nevertheless the reduction in recharge makes in some years the salinity to get close to the one obtained without LULC changes (BL scenario), being very similar at the end of the period. Again CC scenarios would increase the variability of the salinity in the simulated period.

We can also identify in the southern part areas where the situation will clearly improve throughout the future horizon contemplated with the proposed scenarios. For example, at observation point 21, in Oropesa area, the salinity would be reduced with the contemplated scenarios, which would be mainly related with the reduction of pumping in this area due to the transformation of irrigated areas to residential land defined in the LULC scenario.

As commented above the expected results without considering GC impacts are likely to be too pessimistic or optimistic, depending on the location. These results can be useful for the authorities in charge of implementing management policies in the Plana de Oropesa Torrablanca. We can use this coupled modeling framework to assess potential effect of adaptation measures to GC by modifying the inputs of the models. Participatory processes including the relevant stakeholders might be essential in the definition of scenarios and successful adaptation measures (Pulido-Velazquez et al., 2015). This modeling framework could be useful in the search for consensus ("shared vision" models) between different stakeholders.

*We have also added the next paragraph respect to the sensitivity of the results to SLR comparing with other previous studies:*

We find in the literature other examples in which the sensitivity of seawater intrusion to the SLR would be low. Chan et al. (2011) obtained this conclusion in a synthetic confined coastal aquifer in which recharge in unchanged;

Rasmussen et al. (2013) obtained the same conclusion for an inland coastal aquifer with minor SLRs. Nevertheless other authors, as Werner and Simmons (2009) showed that in unconfined aquifers the influence of the inland boundary condition can be significant to its sensitivity to SLR.

**16) Uncertainty: The uncertainty of the results are not touched at all. Considering the chain of model component that are used the total uncertainty of the obtained results must be significant. A discussion of this element is mandatory. Quantification would be even better.**

*We think that we made a mistake including the word uncertainty in the title and we propose to remove it, because it could produce misunderstand about the target of the paper. We agree with the reviewer that a deeper and broader treatment of the uncertainty would be advisable. However, we consider this is out of the scope of the present paper. Note that it would require to deal with different sources of uncertainty. The complexity is even greater for the presented methodology, since it entails the coupling of several numerical codes and a large amount of data and a long simulation time period.*

*There are numerous classification schemes for sources of uncertainty in the literature. In this sense, the uncertainties covered in this work could be summarized as (Matott et al., 2009):*
*-Parameter, model, and modeller uncertainty.*
*-Initial system state, parameter, input, and output uncertainty.*
*-Context, input, parameter, structural, and technical uncertainty.*
*-Statistical variation, subjective judgment, linguistic imprecision, variability, inherent randomness, disagreement, approximation.*

*There are also a lot of quantitative methods and tools for uncertainty assessment in integrated models (Matott et al., 2009), which would be worth a paper by itself when applied to the Plana Oropesa-Torreblanca aquifer.*
*Data analysis (DA): to evaluate or summarize input, response, or model output data.*
*Identifiability analysis (IA): to expose inadequacies in the data or suggest improvements in the model structure.*
*Parameter estimation (PE): to quantify uncertain model parameters using model simulations and available response data.*
*Uncertainty analysis (UA): to quantify output uncertainty by propagating sources of uncertainty through the model.*
*Sensitivity analysis (SA): to determine which inputs are most significant screening, local, global.*
*Multimodel analysis (MMA): to evaluate model uncertainty or generate ensemble predictions.*
*Bayesian networks (BN): to combine prior distributions of uncertainty with general knowledge and site-specific data to yield an updated (posterior) set of distributions.*

*As a further research we could apply some of these models and techniques to deal with the uncertainty.*

*Following the reviewers' comment this consideration has been added to the manuscript within the limitation section.*

*- Matott, L. S., J. E. Babendreier, and S. T. Purucker (2009), Evaluating uncertainty in integrated environmental models: A review of concepts and tools, Water Resour. Res., 45, W06421, doi:10.1029/2008WR007301.*

[revised manuscript text omitted]

**FRANCE**

**PORTUGAL**

**SPAIN**

Irta Mountain

Torreblanca

Estopet

Southern
Maestrazgo

II-II´

Prat de
Cabanes

I-I´

Chinchilla

Mediterranean
Sea

Oropesa
Mountain

Oropesa

Desert de
Les Palmes

0     2 Km.

**Cross Section I-I´**

**Cross Section II-II´**

**Legend**

■ **Village**

■ **Wetland (Prat de Cabanes)**

— **River**

**Mountains**

- - - **Geological cross section**

■ **Mediterranean Sea**

**Geologic Time Scale**

**Plioquaternary**

**Miocene**

**Cretaceous**

**Lower Cretaceous-Jurassic**

**Jurassic**

[Figure]

**Figure 1: Location map area and cross sections of the study.**

[revised manuscript text omitted]

---

## Author Response (AR2)

**Journal: Hydrol. Earth Syst. Sci.**
**Manuscript Number: hess-2017-262**
**Title: "Integrated assessment of future potential global change scenarios and their hydrological impacts in coastal aquifers. A new tool to analyse management alternatives in the Plana Oropesa-Torreblanca aquifer"**
**Special Issue: Assessing impacts and adaptation to global change in water resource systems depending on natural storage from groundwater and/or snowpacks**

**\*\*\*\*\*\*\*\*\*\*\*\*\*\*\*\*\*\*\*\*\*\*\*\*\*\*\*\*\*\*\*\*\*\*\*\*\*\*\*\*\*\*\*\*\*\*\*\*\*\*\*\*\*\*\*\*\*\*\*\*\*\*\*\*\*\*\*\*\***

**Comments from the editors:**
**\*\*\*\*\*\*\*\*\*\*\*\*\*\*\*\*\*\*\*\*\*\*\*\*\*\*\*\*\*\*\*\*\*\*\*\*\*\*\*\*\*\*\*\*\*\*\*\*\*\*\*\*\*\*\*\*\*\*\*\*\*\*\*\*\*\*\*\*\***

**Editor Decision: Publish subject to technical corrections (01 May 2018) by Jesús Carrera**
**Comments to the Author:**
**A paper like this (very broad in scope) can always be subject to criticism for lack of details on specific issues. The authors have responded satisfactorily to the issues that the referees chose to raise.**
**Unfortunately, sometimes such an extensive review causes authors to loose perspective. Therefore, I'd like to invite the authors to have a (non-complusory) re-read of their paper to ensure internal consistency.**
**Specifically,**
**1) You may want to add a perspective paragraph at the end pointing which findings of your work may be of general interest (i,.e., beyond the Plana Oropesa-Torreblanca aquifer)**
**2) You may want to delete the last statement (low sensitivity of heads to SLR) because it is so trivial that it detracts from the rest of the work.**
**3) Km2 in the abstract should be km2**

**Response to Comments from the Editors:**
*We thank the editor comments. Following his suggestions, we have re-read the paper and we have introduced some changes (see them in the R2_Manuscript_changes_marked_Pulido-V_et_al.docx) to improve the clarity of the exposition and to ensure internal consistency.*

*Specifically:*

*1) We have added a perspective paragraph at the end pointing which findings of your work may be of general interest (i,.e., beyond the Plana Oropesa-Torreblanca aquifer).*

In section 4, Results and discussions:

The methodology proposed in this paper to perform an integrated analysis of future potential GC scenarios (considering, LULC, CC and SLR) and their hydrological impacts is general. It can be used to assess the potential status of any coastal aquifer in terms of flow balance components, hydraulic head, and salinity. It includes the definition of future CC scenarios by using equi-feasible and non-equifeasible ensemble of projections based on the results of a multi-criteria analysis of the series generated from several Regional Climatic Models with different downscaling approaches. A modelling framework was proposed to assess hydrological impacts of future climatic scenarios on the coastal aquifer based on a density-dependent simulation whose inputs are defined by sequential coupling of rainfall-recharge models, crop irrigations requirements and

irrigation returns models (a chain of models). This chain of models, calibrated using the available historical data, allow testing the conceptual approximation of the aquifer behavior and the propagation of the generated potential future scenarios. The application of the proposed method allows to improve our knowledge about the case study and so comprise a tool to support decisions about sustainable adaptation management strategies. In the definition of scenarios and plausible adaptation measures to be analyzed participatory processes including the relevant stakeholders might be essential (Pulido-Velazquez et al., 2015). This modeling framework could be useful in the search for consensus ("shared vision" models) between different stakeholders in the definition of the cited scenarios. The limitations and potential future extensions of this work, for example in terms of analyses of uncertainties, are discussed in the next subsection (4.1).

In section 5, Conclusions:

We have proposed a general method to perform an integrated analysis of the potential impacts of future CC, LULC change and SLR in a coastal aquifer. It is assessed in terms of flow balance components, hydraulic head, and salinity distribution. It includes the definition of future CC scenarios by using equi-feasible and non-equifeasible ensemble of projections based on the results of a multi-criteria analysis of the series generated from several Regional Climatic Models with different downscaling approaches. A modelling framework, defined with a chain of models was proposed to assess hydrological impacts of these future climatic scenarios.

*2) Following the editor suggestion, we have deleted the last statement about the low sensitivity of heads to SLR.*

*3) We have corrected the typo error introduced in $km^2$*

[revised manuscript text omitted]